# Teaching Large Language Models to Self-Debug

**Xinyun Chen**[1]   **Maxwell Lin**[2]   **Nathanael Schärli**[1]   **Denny Zhou**[1]

[1] Google DeepMind   [2] UC Berkeley

`{xinyunchen,schaerli,dennyzhou}@google.com, mxlin@berkeley.edu`

## Abstract

Large language models (LLMs) have achieved impressive performance on code generation. However, for complex programming tasks, generating the correct solution in one go becomes challenging. In this work, we propose Self-Debugging, which teaches an LLM to debug its predicted program. In particular, we demonstrate that Self-Debugging can teach the LLM to perform *rubber duck debugging*; i.e., without any human feedback on the code correctness or error messages, the model is able to identify its mistakes by investigating the execution results and explaining the generated code in natural language. Self-Debugging achieves the state-of-the-art performance on several code generation benchmarks, including Spider for text-to-SQL generation, TransCoder for C++-to-Python translation, and MBPP for text-to-Python generation. On the Spider benchmark where there are no unit tests to verify the code correctness, Self-Debugging with code explanation consistently improves the baseline by $2-3\%$, and improves the prediction accuracy on problems of the hardest level by $9\%$. On TransCoder and MBPP where unit tests are available, Self-Debugging improves the baseline accuracy by up to $12\%$. Meanwhile, by leveraging feedback messages and reusing failed predictions, Self-Debugging notably improves sample efficiency, and can match or outperform baseline approaches that generate more than $10\times$ candidate programs.

## 1 Introduction

Code generation has been a long-standing challenge with a variety of applications (Yu et al., 2018; Austin et al., 2021; Li et al., 2022; Roziere et al., 2020). In particular, recent large language models have demonstrated a significant leap in improvement over prior deep neural networks (Chen et al., 2021a; Nijkamp et al., 2023; Zheng et al., 2023; Xu et al., 2022). However, for many programming tasks, generating correct code with a single attempt is challenging. Inspired by observations that correct code is much more likely to be predicted when multiple programs are sampled from the model (Chen et al., 2021a; Chowdhery et al., 2022; Li et al., 2022), one line of work has designed reranking techniques to select the best candidate from multiple samples, which typically requires tens of samples to start with (Shi et al., 2022; Zhang et al., 2022; Ni et al., 2023; Li et al., 2022).

Intuitively, even for human programmers, there is no guarantee that the code written on the first try is always accurate. Instead of completely discarding the incorrect code, humans typically look into the code and investigate execution results, then make changes to resolve the implementation errors. While some recent works show that large language models have potential for generating feedback messages to critique and refine their outputs for some natural language and reasoning domains (Shinn et al., 2023; Madaan et al., 2023b; Kim et al., 2023; Nair et al., 2023; Bai et al., 2022), prior works suggest that such large language models are not yet capable of correcting code when lacking external feedback, such as unit tests or human instructions (Chen et al., 2023a).

In this work, we propose Self-Debugging, where we teach the large language model to debug its own predicted code via few-shot prompting. Without any additional model training, Self-Debugging instructs the model to execute the code, then generate a feedback message based on the code and its execution result. Different from prior works on utilizing human feedback for code repair, where the feedback message explains the code errors and how to fix them (Chen et al., 2023a; Austin et al., 2021), Self-Debugging teaches the model to identify the implementation errors

via investigating into execution results and explaining the code by itself. This debugging process is reminiscent of *rubber duck debugging* for human programmers, where explaining the code line-by-line in natural language to a rubber duck significantly boosts debugging efficiency without expert guidance (Hunt & Thomas, 2000). Figure 1 illustrates the full procedure of SELF-DEBUGGING.

We evaluate SELF-DEBUGGING on a variety of models, showing that SELF-DEBUGGING achieves the state-of-the-art performance on different types of code generation tasks. On the Spider benchmark (Yu et al., 2018) for text-to-SQL generation where there are no unit tests in the problem description, with code explanation, SELF-DEBUGGING consistently improves the baseline by $2 - 3\%$ with different numbers of initial programs, and improves the prediction accuracy on the most complicated SQL queries by $9\%$. On both TransCoder for code translation (Roziere et al., 2020) and MBPP for text-to-Python generation (Austin et al., 2021), utilizing unit tests along with code explanation boosts the accuracy by up to $12\%$, and code explanation alone without debugging also consistently improves the code translation performance by $2 - 3\%$. Meanwhile, SELF-DEBUGGING improves sample efficiency, and can match or outperform baseline models that sample more than $10\times$ predictions. Our work indicates that besides improving their ability to generate code from scratch, teaching large language models to perform SELF-DEBUGGING without human guidance is another promising path to enhance coding capability and reduce the sampling cost required to accomplish challenging tasks.

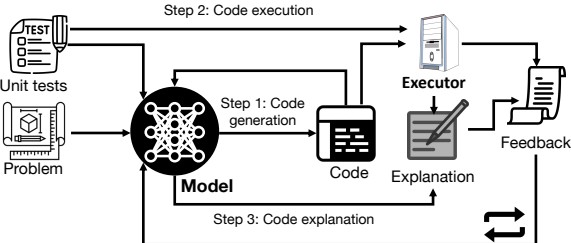

Figure 1: SELF-DEBUGGING with a large language model. At each debugging step, the model first generates new code, then the code is executed and the model explains the code. The code explanation along with the execution results constitute the feedback message, based on which the model infers the code correctness and then adds this message to the feedback. The feedback message is then sent back to the model to perform more debugging steps. When unit tests are not available, the feedback can be purely based on code explanation.

## 2 PROMPTING FOR CODE GENERATION

In this section, we discuss the background on prompting for code generation using large language models. We first introduce few-shot prompting, then discuss how to select the final prediction from multiple samples based on code execution.

**Few-shot prompting.** Few-shot prompting aims to instruct the language model to solve a task with several input-output demonstrations (Brown et al., 2020). Taking text-to-SQL generation as an example, the few-shot prompt prepends the question of interest with a list of (question, SQL) pairs, so that when the model is asked to predict subsequent tokens of the given prompt, it will follow the prompt format to generate the SQL query. Besides input-output demonstrations, we can optionally add an instruction in the prompt to provide a high-level task description (Ouyang et al., 2022; Sanh et al., 2022; Suzgun et al., 2022). For example, in the first two steps of our SELF-DEBUGGING prompt shown in Figure 3, both prompts start with instructions that ask the model to generate explanations. We provide the full few-shot prompts in the appendix.

**Execution-based code selection.** Prior works demonstrate that decoding multiple samples can significantly improve the performance of large language models (Wang et al., 2023; Shi et al., 2022). In particular, for code generation tasks, we can utilize code execution to select the final prediction (Chen et al., 2019; Li et al., 2022; Shi et al., 2022; Zhang et al., 2022; Ni et al., 2023). One line of work selects the final prediction using the majority vote of execution results (Chen et al., 2019; Li et al., 2022; Shi et al., 2022), while other works design reranking schemes to improve the performance (Zhang et al., 2022; Ni et al., 2023; Yin & Neubig, 2019; Zeng et al., 2022). In this work, when there are multiple predictions, we follow the first line of work to select the predicted code

with the most frequent execution result among those that do not encounter execution errors, then apply SELF-DEBUGGING to the code.

Some code generation tasks are accompanied with unit tests to specify the program execution behavior (Chen et al., 2021a; Austin et al., 2021; Li et al., 2022; Hendrycks et al., 2021). Specifically, the unit tests are a set of input-output pairs $\{(i_k, o_k)\}_{k=1}^K$, and a program $P$ passes unit tests when $P(i_k) = o_k, \forall k \in \{1, ..., K\}$. When unit tests are presented in the problem description, we filter out programs that do not pass the unit tests before performing the execution-based majority vote.

## 3 SELF-DEBUGGING FRAMEWORK

Figure 1 illustrates our SELF-DEBUGGING framework for iterative debugging, where we utilize a pretrained large language model without finetuning it. One turn of SELF-DEBUGGING consists of 3 steps: Generation, Explanation, and Feedback.

- For the Generation step, given the problem description, the model predicts candidate programs.
- During the Explanation step, the model is prompted to process the predictions in a semantically useful way, such as explaining the prediction in natural language, or creating an execution trace of the predicted code for a sample input.
- Finally, for the Feedback step, a feedback message concerning the code correctness is generated. This can be determined by asking the model itself, or can be generated externally from unit tests.

The debugging process terminates when the feedback message states that the prediction is correct, or a maximum allowed number of debugging turns is reached.

In practice, a SELF-DEBUGGING turn does not always include all steps above. Figure 2 presents SELF-DEBUGGING feedback formats that can be automatically acquired and generated.

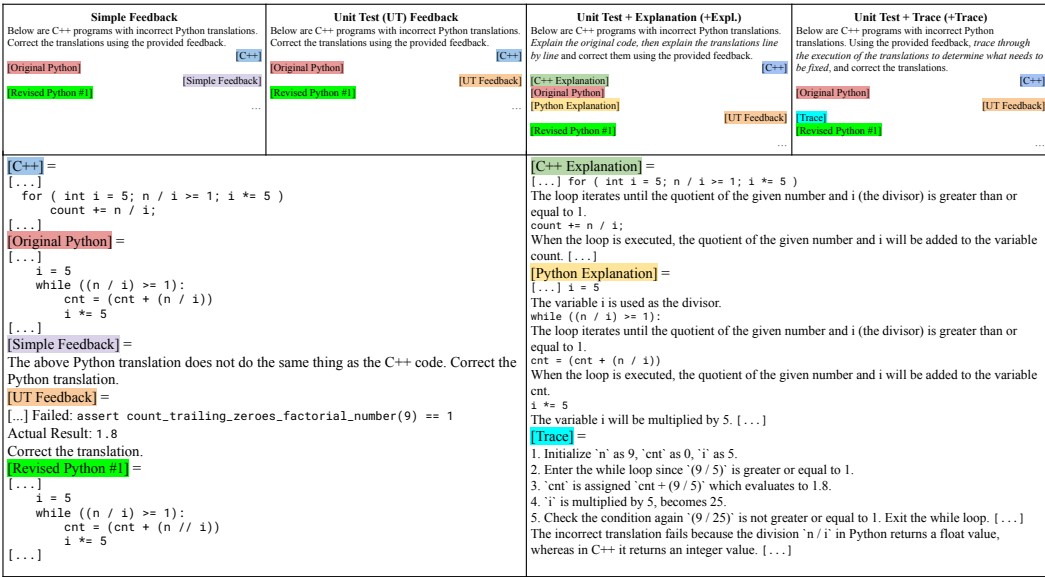

Figure 2: Examples of SELF-DEBUGGING prompts with different feedback formats for code translation. See Appendix F for more explanations.

**Simple feedback.** The simplest form of automatic feedback is a sentence that just indicates the code correctness without more detailed information, which omits the Explanation step in a full SELF-DEBUGGING turn.

**Unit test feedback (UT).** When the problem description includes unit tests, besides utilizing code execution to check code correctness, we can also incorporate the execution results in the feedback, which provides richer information for debugging. Figure 2 presents a sample unit test feedback message. Intuitively, inspecting runtime error messages and execution results of failed unit tests also

helps human programmers debug more effectively. In our experiments, we will demonstrate that leveraging unit tests whenever available consistently boosts the SELF-DEBUGGING performance.

**Code Explanation feedback (Expl).** Despite recent promising progress showing that large language models can generate critiques to avoid harmful model outputs (Ganguli et al., 2023; Bai et al., 2022) and improve their performance on some natural language and reasoning tasks (Shinn et al., 2023; Kim et al., 2023; Saunders et al., 2022), prior work has yet to show the effectiveness of model-generated feedback on code generation (Chen et al., 2023a). On the other hand, large language models have been shown to be capable of describing their generated problem solutions in both text (Wei et al., 2022; Kojima et al., 2022; Zhou et al., 2023) and code (Gao et al., 2022; Chen et al., 2022) formats.

Inspired by these observations, instead of teaching the large language model to predict error messages, we propose SELF-DEBUGGING via explaining the generated code. This debugging process is reminiscent of *rubber duck debugging*, where a programmer debugs code by explaining it line-by-line to a rubber duck (Hunt & Thomas, 2000). By describing the code implementation and comparing it to the problem description, human programmers are usually able to identify the bugs without extra guidance. Empirically, we observe that a large language model can also benefit from rubber duck debugging, especially when unit tests are not available.

**Execution trace feedback (Trace).** Besides explaining the code itself, human programmers also often understand the code semantic meaning by simulating the execution process. Prior work on code repair has demonstrated that training the repair model on execution traces improves the debugging performance (Wang et al., 2018; Gupta et al., 2020). Therefore, when unit tests are available, we examine another explanation feedback format where the LLM is instructed to explain the intermediate execution steps line-by-line. Note that both the execution trace and the line-by-line explanation come from model generation instead of code execution, thus the trace feedback does not require more information than the pure code explanation feedback; i.e., no access to intermediate execution states.

## 4 APPLICATIONS

In our evaluation, we showcase applications of SELF-DEBUGGING to different code generation domains: text-to-SQL generation, code translation, and text-to-Python generation. First, text-to-SQL generation represents the scenario where no unit tests are available, where SELF-DEBUGGING enables the model to identify errors and justify code correctness by explaining the predicted code. In code translation, we demonstrate how to utilize the execution of unit tests to provide feedback messages. Finally, the text-to-Python domain requires the model to infer prediction correctness when only a subset of unit tests are presented in the problem description.

### 4.1 TEXT-TO-SQL GENERATION

The goal of text-to-SQL tasks is to generate the corresponding SQL query given a question and the database information. We evaluate SELF-DEBUGGING on the development set of the Spider benchmark (Yu et al., 2018) where the unit tests are not available, thus it becomes challenging for the model to infer the correctness of the predicted SQL queries.

Figure 3 demonstrates an example to illustrate the SELF-DEBUGGING process. At a high level, the debugging process contains 3 steps. First, we prompt the model to summarize the question and infer the return type required by the question; i.e., the number of table columns in the SQL output. In the second step, the model generates the SQL explanation. Finally, the model compares the inferred SQL explanation and question description, and then predicts the correctness of the current SQL query.

### 4.2 CODE TRANSLATION

Next, we apply SELF-DEBUGGING to code translation, where the goal is to translate code in one programming language into another language. In our experiments, we use the TransCoder dataset (Roziere et al., 2020), which includes a test set of parallel functions in different programming languages along with unit tests. Following (Chowdhery et al., 2022), we evaluate SELF-DEBUGGING on the task of translating C++ code to Python code, with 560 problems and 10 unit tests per problem. The problem description includes both the C++ code and all unit tests.

---

**Step 1: Question explanation**

Infer the return type of the question.

[Question]

Which customers have both "On Road" and "Shipped" as order status? List the customer names.

[Question Explanation]

…The question returns the customer names who have both "On Road" and "Shipped" as order status. So the question returns 1 column.

---

**Step 2: Code explanation**

Summarize the return type of the execution.

[SQL]

SELECT customers.customer_name FROM customers JOIN orders ON customers.customer_id = orders.customer_id WHERE orders.order_status = "On Road" AND orders.order_status = "Shipped"

[SQL Explanation]

…the SQL query returns an empty table, because the "WHERE" clause contains mutually exclusive conditions.

---

**Step 3: Self-Debugging with explanation**

[Question]

[Explanation summary]

As in your explanation, the SQL query returns an empty table, because the "WHERE" clause contains mutually exclusive conditions. The question returns 1 column, the customer names who have order status of both "On Road" and "Shipped".

[Feedback]

So the SQL prediction above is wrong. Please fix the SQL.

[Revised SQL]

SELECT customers.customer_name FROM customers JOIN orders ON customers.customer_id = orders.customer_id WHERE orders.order_status = "On Road" INTERSECT SELECT customers.customer_name FROM customers JOIN orders ON customers.customer_id = orders.customer_id WHERE orders.order_status = "Shipped"

[Explanation summary #1]

As in your explanation, the SQL query returns a table with 1 column, the customer names who have the order status "On Road" and "Shipped". The question returns 1 column, the customer names who have order status of both "On Road" and "Shipped". So the question returns 1 column.

[Feedback #1]

So the SQL prediction above is correct!

---

Figure 3: An example of SELF-DEBUGGING prompting for text-to-SQL generation. The model is required to predict the code, explanation and feedback. See Appendix F for more explanations.

Given the availability of unit tests, we apply SELF-DEBUGGING only when the predicted Python code does not pass all the unit tests. Figure 2 provides sample SELF-DEBUGGING prompts.

### 4.3 TEXT-TO-PYTHON GENERATION

In addition, we evaluate SELF-DEBUGGING in the setting where only a subset of unit tests are presented in the problem description. Specifically, we perform experiments on the test set of MBPP (Austin et al., 2021), which contains 500 Python problems with text descriptions, where each problem has 3 unit tests. We follow prior work (Shi et al., 2022; Ni et al., 2023) in including the first unit test in the prompt as part of the problem description, and keeping the remaining 2 unit tests hidden for full evaluation. Similar to code translation, we can also utilize the unit test execution results in the feedback message, but the main difference is that the model still needs to infer the code correctness even if the predicted Python code passes the given unit test.

## 5 EXPERIMENTS

We evaluate SELF-DEBUGGING on `code-davinci-002` ( Chen et al. (2021a), referred to as Codex), `gpt-3.5-turbo`, `gpt-4` (OpenAI, 2023), and StarCoder (Li et al., 2023b) with 15.5B parameters. For initial code generation, when starting from one program, we perform greedy decoding with temperature $\tau = 0$. When sampling multiple programs for a problem, we set temperature $\tau = 0.7$, then we perform execution-based selection described in Section 2. For Transcoder and MBPP, we also evaluated an oracle baseline, which has access to all unit tests and samples another program when the greedy decoding baseline makes the wrong prediction. All experiments for SELF-DEBUGGING use greedy decoding to generate feedback messages and new programs. We set

the maximum number of debugging turns to be 10, though empirically the successful debugging processes mostly end within 3 turns. We present the full prompts for experiments in the appendix.

We evaluate SELF-DEBUGGING against two types of code reranking baselines as follows.

**Models trained for the given task.** The Spider benchmark contains a training set of over 10K samples, and the state-of-the-art models are mostly finetuned on this training set. We compare SELF-DEBUGGING to T5-3B + N-best Reranking (Zeng et al., 2022), where the T5-3B model is specially trained for text-to-SQL generation. Although LEVER (Ni et al., 2023) also utilizes Codex to generate candidate SQL queries, they train a verifier to select the final prediction based on execution, and thus this approach also requires extra training. For both TransCoder and MBPP benchmarks, the state-of-the-art results are all accomplished by LLMs, thus we defer the comparison to Appendix D.

**Prompting-based approaches.** We compare SELF-DEBUGGING against recent approaches that also only perform prompting without any additional training. Specifically, Both MBR-Exec (Shi et al., 2022) and Coder-Reviewer (Zhang et al., 2022) first generate multiple candidate programs by prompting the pretrained model. Afterward, MBR-Exec (Shi et al., 2022) selects the program with the most common execution output, while Coder-Reviewer (Zhang et al., 2022) selects the program by utilizing both the likelihood of the predicted code given the problem description (Coder score) and the likelihood of the problem description given the predicted code (Reviewer score).

## 5.1 MAIN RESULTS

Table 1: Comparing SELF-DEBUGGING to prior ranking techniques. The baseline refers to our best results without SELF-DEBUGGING, and uses multiple samples.

(a) Results on the Spider development set.

| | Spider (Dev) |
| --- | --- |
| *w/ training* | |
| T5-3B + N-best Reranking | 80.6 ($n = 25$) |
| LEVER (Ni et al., 2023) | 81.9 ($n = 50$) |
| *Prompting only w/o debugging* | |
| Coder-Reviewer | 74.5 ($n = 25$) |
| MBR-Exec | 75.2 ($n = 50$) |
| SELF-DEBUGGING (this work) | |
| Codex (baseline) | 81.3 ($n = 32$) |
| + Expl. | **84.1** |

(b) Results on MBPP dataset.

| | $n$ samples |
| --- | --- |
| Prior work (Prompting with reranking) | |
| MBR-Exec | 63.0 ($n = 25$) |
| Reviewer | 66.9 ($n = 25$) |
| LEVER | 68.9 ($n = 100$) |
| SELF-DEBUGGING (this work) | |
| Codex (baseline) | 72.2 ($n = 10$) |
| Simple | 73.6 |
| UT | 75.2 |
| UT + Expl. | **75.6** |

First, we compare SELF-DEBUGGING to prior code reranking approaches in Table 1, where both SELF-DEBUGGING and prior prompting-based approaches use Codex. We demonstrate that SELF-DEBUGGING consistently improves the performance.

Next, we compare different feedback formats for SELF-DEBUGGING with different LLMs in Table 2. On the Spider benchmark where unit tests are not available, simple feedback alone does not notably improve the performance, because the model typically struggles to distinguish between correct and wrong SQL queries without explanation, and does not produce meaningful changes to the initial predicted SQL queries. Note that on TransCoder and MBPP benchmarks where at least one unit test is available for SELF-DEBUGGING, simple feedback also utilizes the execution result to infer the code correctness even if the execution information is not presented in the feedback message, thus simple feedback still improves the model prediction for both applications. In Section 5.2.2, we further present an ablation study showing that leveraging code execution is crucial for enabling a performance leap with SELF-DEBUGGING. For all tasks, models generally benefit from richer feedback for SELF-DEBUGGING, especially when the execution information is present in the feedback. The oracle baseline results further justify that SELF-DEBUGGING is more effective than completely discarding wrong predictions and sampling another program independently. Note that on MBPP, the oracle baseline has access to all unit tests to determine the code correctness, while SELF-DEBUGGING only has access to 1 unit test, as described in Section 4.3. Nevertheless, the oracle baseline still generally performs worse than SELF-DEBUGGING with the simple feedback, let alone other feedback types.

Table 2: Results of SELF-DEBUGGING with different feedback formats. The baseline corresponds to the greedy decoding performance. Note that the oracle baseline generates another sample when the greedy decoding prediction is wrong, assuming access to all unit tests.

(a) Results on the Spider development set.

| Spider | Codex | GPT-3.5 | GPT-4 | StarCoder |
|---|---|---|---|---|
| Baseline | 77.5 | 71.1 | 73.2 | 64.7 |
| Simple | 77.5 | **72.2** | 73.4 | **64.9** |
| +Expl. | **80.8** | **72.2** | **73.6** | **64.9** |

(b) Results on TransCoder.

| TransCoder | Codex | GPT-3.5 | GPT-4 |
|---|---|---|---|
| Baseline | 80.4 | 89.1 | 77.3 |
| Oracle Baseline | 84.5 | 91.3 | 80.0 |
| Simple | 89.3 | 91.6 | 80.9 |
| UT | 91.6 | **92.7** | 88.8 |
| + Expl. | **92.5** | **92.7** | **90.4** |
| + Trace. | 87.9 | 92.3 | 89.5 |

(c) Results on MBPP.

| MBPP | Codex | GPT-3.5 | GPT-4 | StarCoder |
|---|---|---|---|---|
| Baseline | 61.4 | 67.6 | 72.8 | 47.2 |
| Oracle Baseline | 65.4 | 72.6 | 77.0 | 50.0 |
| Simple | 68.2 | 70.8 | 78.8 | 50.6 |
| UT | 69.4 | 72.2 | **80.6** | 52.2 |
| + Expl. | 69.8 | **74.2** | 80.4 | 52.2 |
| + Trace. | **70.8** | 72.8 | 80.2 | **53.2** |

By comparing different LLMs, we observe that:

- Although GPT-4 is shown to be much stronger than previous GPT models on many tasks (OpenAI, 2023), on Spider, both its initial SQL generation and SELF-DEBUGGING performance are much worse than Codex. One possible reason is that GPT-4 is tuned for zero-shot prompting, and thus it does not follow few-shot exemplars as well as Codex. Meanwhile, both GPT-3.5 and GPT-4 might not be optimized for SQL generation, and thus their zero-shot results are more than 10% worse than their few-shot counterparts.

- GPT-4 is significantly better than Codex and GPT-3.5 for initial Python code generation on MBPP. Meanwhile, its self-debugging performance gain is on par with Codex: over 12% improvement on Transcoder, and 8% improvement on MBPP.

- Despite that the baseline performance of StarCoder is considerably worse than GPT models, SELF-DEBUGGING with unit test execution also offers a significant performance gain, i.e., 6% on MBPP with the execution trace feedback.

## 5.2 ABLATION STUDIES

### 5.2.1 SELF-DEBUGGING IMPROVES THE SAMPLE EFFICIENCY

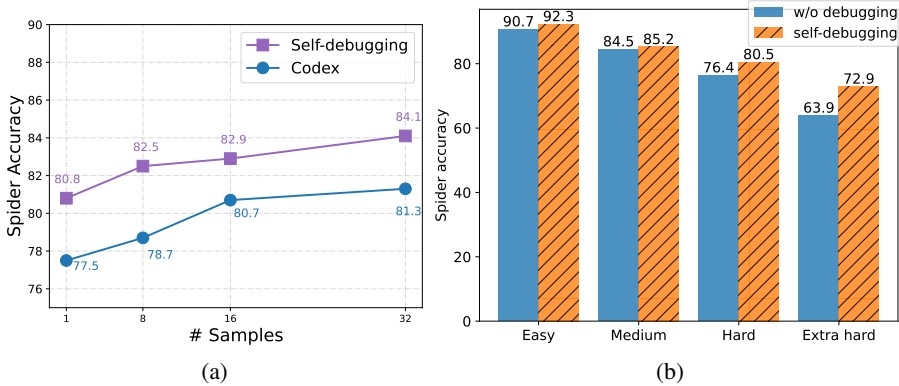

(a)                                    (b)

Figure 4: Ablation studies on the Spider development set with Codex. (a) Accuracies with different numbers of initial samples. (b) Breakdown accuracies on problems with different hardness levels.

Figure 4a demonstrates that SELF-DEBUGGING notably improves the sample efficiency with different numbers of initial samples. In particular, on Spider, applying SELF-DEBUGGING to predictions gener-

ated with greedy decoding matches the baseline accuracy using 16 samples, and SELF-DEBUGGING from 8 samples outperforms the baseline accuracy using 32 samples. Note that typically one debugging turn is sufficient, and the accuracy improvement after one turn is within $0.1\%$. Appendix E presents similar sample efficiency improvement on other benchmarks.

### 5.2.2 IMPORTANCE OF CODE EXECUTION

Table 3: Results of SELF-DEBUGGING without unit test execution.

<table>
<tr><td colspan="4">(a) Results on Transcoder.</td><td colspan="4">(b) Results on MBPP</td></tr>
<tr><td>TransCoder</td><td>Codex</td><td>GPT-3.5</td><td>GPT-4</td><td>MBPP</td><td>Codex</td><td>GPT-3.5</td><td>GPT-4</td></tr>
<tr><td>Baseline</td><td>80.4</td><td>**89.1**</td><td>77.3</td><td>Baseline</td><td>61.4</td><td>67.6</td><td>72.8</td></tr>
<tr><td>Simple</td><td>83.4</td><td>**89.1**</td><td>78.2</td><td>Simple</td><td>57.6</td><td>68.2</td><td>76.0</td></tr>
<tr><td>+ Expl.</td><td>**83.9**</td><td>**89.1**</td><td>78.0</td><td>+ Expl.</td><td>64.4</td><td>68.2</td><td>76.0</td></tr>
<tr><td>+ Trace.</td><td>**83.9**</td><td>**89.1**</td><td>**78.4**</td><td>+ Trace.</td><td>**66.2**</td><td>**69.2**</td><td>**76.4**</td></tr>
</table>

By default, we leverage unit test execution for SELF-DEBUGGING when applicable. In Table 3, we examine the scenario where the SELF-DEBUGGING process does not involve code execution on Transcoder and MBPP, thus models need to fully rely on themselves to infer the code correctness as in Spider experiments. Our main findings are as follows:

- With Codex, SELF-DEBUGGING still improves the performance by up to 5%, and the execution trace feedback consistently improves over the simple feedback performance.
- GPT-4 without unit test execution improves the MBPP accuracy by 3.6%, and the improvement on other benchmarks is up to around 1%.
- Compared to Codex, few-shot prompting does not notably improve the SELF-DEBUGGING performance of GPT-3.5 and GPT-4. Without unit test execution, both models tend to be overconfident in their own initial predictions, while GPT-4 is better on Python generation.

In summary, code execution plays an important role in SELF-DEBUGGING; however, sometimes LLMs can still improve their performance purely with self-generated feedback. In particular, execution trace feedback offers consistent improvement.

### 5.2.3 ERROR TYPES FIXED BY SELF-DEBUGGING

To further understand how SELF-DEBUGGING improves performance, we first measure the breakdown accuracies on Spider problems with different difficulty levels, where the difficulty is annotated based on the complexity of the ground truth SQL queries. Figure 4b demonstrates that the improvement achieved by SELF-DEBUGGING is more significant on hard problems. In particular, on extra hard problems, SELF-DEBUGGING increases the accuracy by $9\%$. Table 4a presents a categorization on error types that are successfully fixed by SELF-DEBUGGING. In general, we observe that the initial SQL queries generated by LLMs are usually not completely wrong, but they tend to make small mistakes when the questions require more complex SQL queries, e.g., missing a few WHERE conditions or SELECT columns. In this case, SELF-DEBUGGING with code explanation facilitates the LLM to identify the discrepancy between the question and the predicted SQL query, resulting in an accuracy boost for more complex tasks. See Figure 10 for a sample fix for an extra hard problem.

On the other hand, on Transcoder and MBPP, 60%-70% successful cases fix the output mismatch error when the initial wrong code is very close to a correct one, as shown in Table 4b. Specifically, on Transcoder, over 30% of successful fixes address implementation differences between different programming languages, where the issue is made more obvious with code execution. On MBPP, while 2/3 of the initial programs have problem-specific semantic errors, over 10% of the initial programs can be fixed by switching the order of function arguments and matching the argument types.

## 6 RELATED WORK

**Code repair.** Program repair is an area concerned with fixing bugs in code, where several neural networks have been developed to handle different types of bugs (Gupta et al., 2017; Yasunaga &

Table 4: Breakdown on percentages of error types fixed by SELF-DEBUGGING.

(a) Breakdown on Spider with `code-davinci-002`.

| Error type | % |
|---|---|
| Wrong WHERE conditions | 25.7 |
| Missing the DISTINCT keyword | 17.1 |
| Wrong JOIN clauses | 14.3 |
| Wrong number of SELECT columns | 11.4 |
| Wrong INTERSECT/UNION clauses | 8.6 |
| Wrong aggregate functions and keywords | 5.8 |
| Wrong COUNT columns | 5.7 |
| Wrong column selection | 5.7 |
| Missing nested conditions | 5.7 |

(b) Breakdown on Transcoder with `gpt-3.5-turbo`, and MBPP with `gpt-4`.

| Error type | Transcoder | MBPP |
|---|---|---|
| Output mismatch | 61.9 | 69.2 |
| Runtime errors | 38.1 | 30.8 |

Liang, 2021; Gupta et al., 2020; Wang et al., 2018; Le et al., 2022). While some methods train repair models that only take the code as input (Gupta et al., 2017; Yasunaga & Liang, 2021), other approaches incorporate additional information including execution traces (Gupta et al., 2020; Wang et al., 2018) as well as compiler and execution errors (Yasunaga & Liang, 2020; Le et al., 2022). Our SELF-DEBUGGING also uses additional information to aid in code repair, including execution results and self-generated code explanations. In contrast to prior work on training a separate code repair model, SELF-DEBUGGING utilizes pretrained LLMs, and teaches the model to debug via prompting.

**Prompting with feedback.** Recent works have shown the great promise of RLHF-trained models to generate critiques with prompting, which reduces harmful model outputs (Bai et al., 2022; Ganguli et al., 2023) and improves the performance on some reasoning tasks (Shinn et al., 2023; Madaan et al., 2023b; Kim et al., 2023; Nair et al., 2023). Reflexion (Shinn et al., 2023) and RCI (Kim et al., 2023) prompt an LLM-powered agent to reflect and improve its actions, which utilize the oracle reward to determine whether another trial is needed. Self-Refine (Madaan et al., 2023b) shows that GPT-3.5 and GPT-4 can iteratively generate feedback and revisions of its outputs to improve the final output quality on several tasks such as text generation. Finally, DERA (Nair et al., 2023) simulates a dialogue between two agents to improve the model's predictions in the clinical domain. In this work, we focus on code generation applications, and SELF-DEBUGGING demonstrates the effectiveness of both self-generated feedback and unit test feedback acquired by code execution. In particular, SELF-DEBUGGING instructs the LLM to notice prediction errors via code explanation in the style of rubber duck debugging, which enables the LLM to fix prediction errors even when the unit tests are not available. We also show that we can teach LLMs to perform SELF-DEBUGGING via few-shot prompting, even if the LLM is not specially tuned for zero-shot debugging.

See Appendix B for an extended discussion of related work.

## 7 CONCLUSION

In this work, we presented SELF-DEBUGGING, which enables LLMs to debug code generated by themselves. In particular, we demonstrate that SELF-DEBUGGING empowers the model to perform *rubber duck debugging*, so that the model can identify and fix bugs without human instructions. SELF-DEBUGGING achieves the state-of-the-art performance across several code generation domains, and notably improves sample efficiency.

Our work highlights the promise of improving the coding performance of LLMs by teaching them to debug their own predictions, instead of requiring the model to generate the correct code from scratch. SELF-DEBUGGING instructs the model to understand the code, identify the errors, and follow the error messages to fix the bugs. We consider improving the model's ability to conduct all these steps as important future work. In particular, we hypothesize that better code explanation ability leads to better debugging performance. One direction is to instruct the model to better describe the high-level semantic meaning of code along with the implementation details in its explanation. Another direction is to include additional debugging information in the model feedback, such as a description of potential bugs. Our preliminary results suggest that model-generated feedback messages about semantic errors do not provide additional benefits on top of line-by-line code explanation, and future work can explore techniques to predict more informative error messages.

## ETHICS STATEMENT

Large language models with strong coding capabilities, combined with SELF-DEBUGGING that allows automated improvement of code quality, may eventually allow for the automation of the development and maintenance of large code bases in addition to the automated explanation of their functionalities and vulnerabilities. Although this can provide great productivity gains, on the flip side, the increasing reliance of model-generated code might also open up new vulnerabilities. Therefore, to deploy any large language model for real-world programming applications, it is crucial to systematically analyze its capabilities and limitations before widely utilizing code generated by the model.

## REPRODUCIBILITY STATEMENT

We run all experiments on public benchmarks. We use publicly accessible large language models for evaluation: API access of GPT models is available at `https://openai.com/api`, and StarCoder is an open-source LLM (Li et al., 2023b). See Appendices H, I and J for full prompts in experiments on Spider, Transcoder and MBPP benchmarks.

## ACKNOWLEDGEMENT

We would like to thank Jacob Austin, Quoc V. Le, Charles Sutton, Hanjun Dai, and Olivier Bousquet for helpful discussion and feedback.

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

## A  LIMITATIONS

Our work demonstrates the ability for large language models to improve code generation quality with self-generated feedback and execution results. Despite the fact that SELF-DEBUGGING achieves significant performance gains on several code generation benchmarks, the performance is ultimately bounded by the capability of the underlying large language model, and there is still a large room for improvement. In particular, real-world programming tasks within large-scale projects can be more complex and include ambiguous task specifications, where both the model-generated feedback and back-and-forth interactions with users are beneficial for improving the coding performance. Meanwhile, how to scale up SELF-DEBUGGING to diagnose issues in large code bases remains an important future direction, where bugs can emerge from interactions between many separate parts of the code.

## B  EXTENDED RELATED WORK

**Language models for code.** Recent years have witnessed rapid progress in deep neural networks for code generation (Devlin et al., 2017; Chen et al., 2019; Yu et al., 2018; Roziere et al., 2020). While models designed and trained for specialized domains have achieved impressive performance in various applications such as text-to-code generation (Li et al., 2023a; Wang et al., 2020; Scholak et al., 2021; Dong & Lapata, 2016; Iyer et al., 2018) and code translation (Chen et al., 2018; Roziere et al., 2020; 2022), latest work on large language models demonstrate that a single pretrained model can achieve the state-of-the-art performance across a wide variety of coding tasks without specialized finetuning (Chen et al., 2021a; Chowdhery et al., 2022; Nijkamp et al., 2023; Zheng et al., 2023; Xu et al., 2022; Athiwaratkun et al., 2023; Orlanski et al., 2023).

Despite showing the remarkable ability to follow natural language instructions, large language models still exhibit limited understanding of code execution (Austin et al., 2021; Li et al., 2022). Specifically, even when the unit tests are provided in the problem description, the generated programs may still violate them (Li et al., 2022; Shi et al., 2022; Ni et al., 2023). Therefore, several approaches have been proposed to leverage code execution to choose the final prediction from multiple candidates, such as utilizing or training a language model for reranking (Zhang et al., 2022; Ni et al., 2023), and performing selection based on the consensus on unit test execution outputs among samples (Chen et al., 2019; 2021b; Roziere et al., 2022; Shi et al., 2022; Li et al., 2022; Chen et al., 2023b). In this work, our main focus is to utilize and explain code execution for SELF-DEBUGGING, which improves the sample efficiency compared to utilizing execution solely for initial code generation.

**Prompting techniques.** Several prompting methods have been proposed to unlock latent abilities of large language models (Wei et al., 2022; Nye et al., 2021; Kojima et al., 2022; Zhou et al., 2023; Khot et al., 2022; Zhou et al., 2022; Gao et al., 2022; Chen et al., 2022; Weng et al., 2022). Rather than prompting a model to directly produce the desired result, these methods have the model first produce useful intermediate outputs. For example, chain-of-thought prompting asks the model to produce intermediate reasoning steps in natural language, which can be accomplished either with few-shot exemplars (Wei et al., 2022) or in a zero-shot manner (Kojima et al., 2022). Meanwhile, several prompting techniques explicitly direct the model to break down a problem into subproblems (Zhou et al., 2023; Khot et al., 2022). For example, decomposed prompting (Khot et al., 2022) delegates subproblems to other LLM instances that are prompted to specialize in specific tasks. Self-verification (Weng et al., 2022) computes verification scores to select the final response given multiple candidate answers, where the backward verification process is primarily designed for arithmetic reasoning tasks. Our prompting format of code explanation is relevant in spirit to chain-of-thought prompting, as the line-by-line code explanation in natural language facilitates analysis of the code that is useful for the debugging task. On the other hand, SELF-DEBUGGING also decomposes the debugging process into several stages, and triggers separate prompts for code explanation and feedback generation.

**Training with feedback.** Training with feedback to improve the outputs of large language models, both in terms of correctness and alignment with human preferences, is an active research direction nowadays (Ziegler et al., 2019; Korbak et al., 2023; Ganguli et al., 2023; Bai et al., 2022). One popular technique is reinforcement learning from human feedback (RLHF) (Ziegler et al., 2019; Ouyang et al., 2022), and RLHF-trained models have demonstrated the ability to avoid harmful outputs when instructed to do so in the prompt (Ganguli et al., 2023). Constitutional AI (Bai et al., 2022) introduces another path toward training harmless models, where they use the pretrained model itself to create automated feedback for both supervised learning and RLHF: for the former, a set of principles are used to guide a language model in creating revisions of its own responses that it is then trained on, and for the latter the same principles are used to prompt a separate model for the feedback needed for RLHF.

Another line of work trains a language model to refine the initial model outputs based on external feedback on prediction quality (Welleck et al., 2023; Liu et al., 2023), which improves the performance on several natural language and reasoning tasks. For code generation, a number of works have trained models to perform code optimization (Madaan et al., 2023a), interact with users for multi-turn code generation (Yu et al., 2019; Yao et al., 2019; Nijkamp et al., 2023), and fix generated programs based on human feedback (Iyer et al., 2017; Elgohary et al., 2020; Chen et al., 2023a; Le et al., 2022). On

the other hand, SELF-DEBUGGING enables the model to generate feedback messages on its own at test time, and does not require extra training.

## C   SELF-DEBUGGING WITH INITIAL SAMPLES FROM DIFFERENT LLMS

Table 5: Results of SELF-DEBUGGING using `gpt-3.5-turbo` (GPT-3.5) and `code-davinci-002` (Codex) on (a) Spider; (b) Transcoder and (c) MBPP. The baseline results are the highest accuracies before SELF-DEBUGGING, which are obtained by Codex for Spider, and GPT-3.5 for Transcoder and MBPP.

| Spider | Codex | GPT-3.5 |
|---|---|---|
| Baseline | 81.3 | |
| Simple | 81.3 | 81.7 |
| +Expl. | **84.1** | 83.0 |

(a)

| TransCoder | Codex | GPT-3.5 |
|---|---|---|
| Baseline | 89.1 | |
| Simple | 90.2 | 91.6 |
| UT | 92.1 | **92.7** |
| UT + Expl. | **92.7** | 92.7 |

(b)

| MBPP | Codex | GPT-3.5 |
|---|---|---|
| Baseline | 67.6 | |
| Simple | 72.4 | 70.8 |
| UT | 73.2 | 72.2 |
| UT + Expl. | 73.6 | **74.2** |

(c)

In Table 5b we compare the results of SELF-DEBUGGING using Codex and GPT-3.5 respectively. For each benchmark, the baseline accuracy presents the best code generation performance obtained by Codex and GPT-3.5. On TransCoder and MBPP, the baseline uses GPT-3.5 with zero-shot prompting. On Spider, we observe that the performance of GPT-3.5 is significantly worse than Codex. For example, when using greedy decoding, GPT-3.5 achieves an accuracy of $59.9\%$ with zero-shot prompting, and $71.1\%$ with few-shot prompting, which is over $6\%$ lower than Codex. On Spider, we utilize the initial code generation from Codex. We present GPT-3.5 SELF-DEBUGGING results using zero-shot prompting, as we observe that the performance with few-shot exemplars is similar. Specifically, we used zero-shot SELF-DEBUGGING prompts which removed exemplars and adapted the instructions from our few-shot prompts to fit the conversation format of GPT-3.5. For example, the instruction included in the simple feedback is *"Does the SQL match the question? If not, generate the fixed SQL."* for Spider.

From Tables 5b and 5c, we show that while GPT-3.5 notably outperforms Codex on generating initial code in Python, the SELF-DEBUGGING performance of Codex is on par with GPT-3.5 on Transcoder. It also performs close to GPT-3.5 on MBPP, and outperforms it for both Simple and UT. In Table 5a, we observe that Codex also outperforms GPT-3.5 on SELF-DEBUGGING for text-to-SQL generation, and code explanation again improves the performance for GPT-3.5. These results demonstrate the effectiveness of our SELF-DEBUGGING prompt with few-shot exemplars, while suggesting that Codex is better than GPT-3.5 at learning from few-shot exemplars with long context.

## D   TRANSCODER COMPARISON TO BASELINE MODELS

Table 6 compares SELF-DEBUGGING to baseline models without debugging.

Table 6: TransCoder dataset (Roziere et al., 2020) for C++ to Python translation.

| | TransCoder (pass@1) |
|---|---|
| *w/o debugging* | |
| TransCoder | 44.5 |
| PaLM | 51.8 |
| PaLM-Coder | 55.1 |
| SELF-DEBUGGING (this work) | |
| Codex | 80.4 |
| UT + Expl. | **92.5** |

## E  MORE DISCUSSION OF SELF-DEBUGGING RESULTS

### E.1  CODE TRANSLATION

For generating initial Python translation, we apply the same few-shot prompt for TransCoder as (Chowdhery et al., 2022), which consists of 3 exemplars (Appendix I.1). From Figure 5a, we again observe that the major improvement comes from the first debugging turn. Specifically, a single debugging turn with the full feedback improves over the greedy decoding accuracy by around 12%. Compared to Figure 5b, applying SELF-DEBUGGING to greedy decoding outperforms the baseline accuracy with 5 samples, and is close to the baseline accuracy with 10 samples.

Meanwhile, incorporating both unit test execution and code explanation improves the debugging performance, and we present some examples in Figures 11 and 12 in Appendix G. In addition, we demonstrate that leveraging code explanation alone without SELF-DEBUGGING also provides a consistent performance gain of $2 - 3\%$ for different numbers of samples, as shown in Figure 5b.

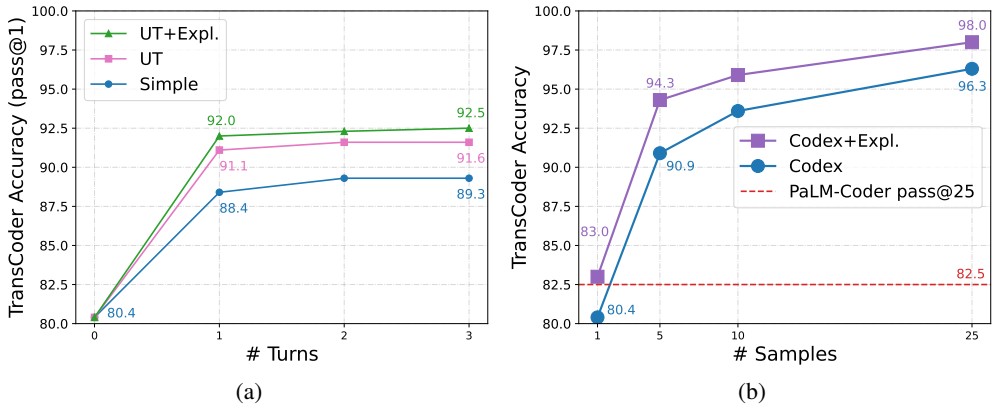

(a)

(b)

Figure 5: Ablation studies on TransCoder. (a) The accuracy of SELF-DEBUGGING prompts with different numbers of debugging turns. (b) The accuracy with different numbers of initial samples. Note that Codex+Expl. only prompts the model to generate explanations for the input C++ code, and does not perform SELF-DEBUGGING.

### E.2  TEXT-TO-PYTHON GENERATION

For initial Python code generation on MBPP, we apply the same 3-shot prompt as (Ni et al., 2023). With greedy decoding, SELF-DEBUGGING increases the accuracy by over 8%, and leveraging both unit test execution and code explanation again consistently improves the performance.

We observe that the improvement using code explanation is less significant on MBPP. First, due to the availability of unit tests, it is much easier to recognize wrong predictions from unit test feedback. Second, due to the context length limit, the number of exemplars in the prompt for SELF-DEBUGGING is smaller when adding code explanation in the feedback message. Specifically, prompts with both simple and unit test feedback messages include 6 exemplars, while the prompt with both the unit test execution and code explanation has 3 exemplars. Third, we find that code explanation plays a smaller role when the initial predicted code is far from the correct solution, in which case the model simply generates a new program and does not rely on code snippets in the initial prediction. See Figure 13 in Appendix G for an example where SELF-DEBUGGING makes major changes to the initial code.

## F  EXAMPLES FOR SPIDER AND TRANSCODER

Figure 6 and 7 illustrate the feedback formats for Spider and Transcoder. The formats of MBPP are similar to TransCoder. See Figures 8 and 9 for examples in the Spider and TransCoder benchmarks.

---

**Step 1: Question explanation**

Infer the return type of the question.

[Question]

Which customers have both "On Road" and "Shipped" as order status? List the customer names.

[Question Explanation]

"List the customer names" returns 1 column. The question returns the customer names who have both "On Road" and "Shipped" as order status. So the question returns 1 column.

---

**Step 2: Code explanation**

Summarize the return type of the execution.

[SQL]

SELECT ustomers.customer_name FROM customers JOIN orders ON customers.customer_id = orders.customer_id
WHERE orders.order_status = "On Road" OR orders.order_status = "Shipped"
Execution:
| George |

[SQL Explanation]

The execution of the SQL query above would return a table with 1 column.
The first column, "customers.customer_name" would contain the customer names.
With "customers JOIN orders", the table would contain the data about customers with orders.
With "WHERE orders.order_status = 'On Road' OR orders.order_status = 'Shipped'", the table filters the records to only include customers who have order status "On Road" or "Shipped".
So the SQL query returns a table with 1 column, the customer names who have the order status "On Road" or "Shipped".

---

**Step 3: Self-Debugging with explanation**

[Question]
[Explanation summary]

As in your explanation, the SQL query returns a table with 1 column, the customer names who have the order status "On Road" or "Shipped". The question returns 1 column, the customer names who have both "On Road" and "Shipped" as order status.

[Feedback]

So the SQL prediction above is wrong. Please fix the SQL.

[Revised SQL #1]

SELECT customers.customer_name FROM customers JOIN orders ON customers.customer_id = orders.customer_id
WHERE orders.order_status = "On Road" AND orders.order_status = "Shipped"

[Explanation summary #1]

As in your explanation, the SQL query returns an empty table, because the "WHERE" clause contains mutually exclusive conditions. The question returns 1 column, the customer names who have order status of both "On Road" and "Shipped".

[Feedback #1]

So the SQL prediction above is wrong. Please fix the SQL.

[Revised SQL #2]

SELECT customers.customer_name FROM customers JOIN orders ON customers.customer_id = orders.customer_id WHERE orders.order_status = "On Road" INTERSECT SELECT customers.customer_name FROM customers JOIN orders ON customers.customer_id = orders.customer_id WHERE orders.order_status = "Shipped"

[Explanation summary #2]

As in your explanation, the SQL query returns a table with 1 column, the customer names who have the order status "On Road" and "Shipped". The question returns 1 column, the customer names who have order status of both "On Road" and "Shipped". So the question returns 1 column.

[Feedback #2]

So the SQL prediction above is correct!

---

Figure 6: An example of SELF-DEBUGGING prompting for text-to-SQL generation. The model is required to predict the code, explanation and feedback. When the returned table has more than 2 rows, only the first 2 rows are included in the prompt. Database information is omitted in the figure for clarity, and we present the full prompts in Appendix H.

## G   EXAMPLES OF MODEL PREDICTIONS WITH SELF-DEBUGGING

See Figures 10, 11, 12 and 13 for examples of model predictions with SELF-DEBUGGING.

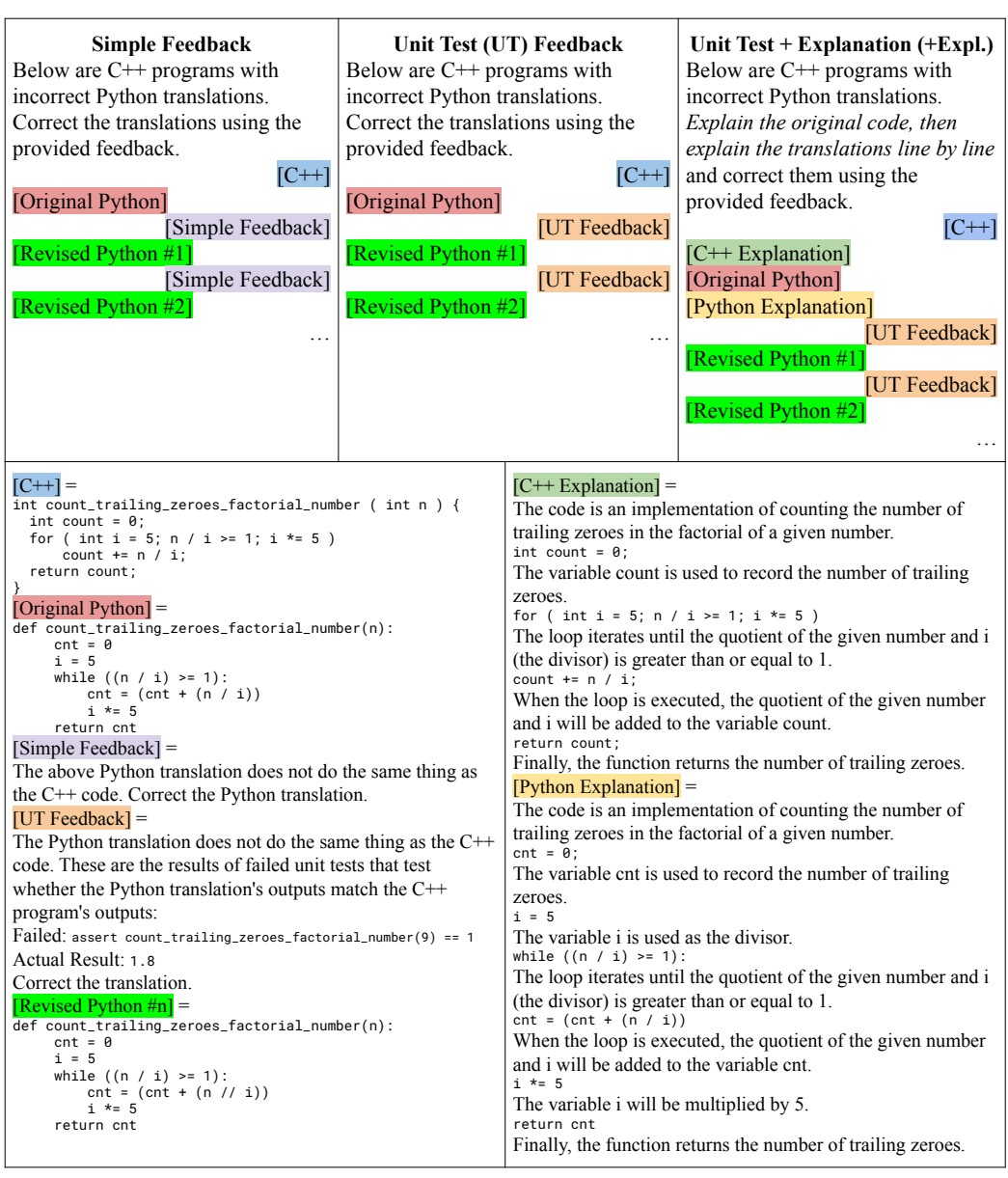

Figure 7: Examples of SELF-DEBUGGING prompts for code translation. Left-aligned blocks are model predictions, and right-aligned blocks contain the input C++ code and feedback messages based on code execution. The full prompts are in Appendix I.

**Problem description**

```
CREATE TABLE customers (
customer_id number ,
customer_name text ,
customer_details text ,
primary key ( customer_id )
)
insert into customers (customer_id, customer_name, customer_details) values (1,
 'Savannah', 'rerum') ;

CREATE TABLE orders (
order_id number ,
customer_id number ,
order_status text ,
date_order_placed time ,
order_details text ,
primary key ( order_id ),
foreign key ( customer_id ) references customers ( customer_id )
)
insert into orders (order_id, customer_id, order_status, date_order_placed,
order_details) values (1, 2, 'Shipped', '2009-02-21 15:26:19', NULL);
...

Question: Which customers have both "On Road" and "Shipped" as order status?
List the customer names.
```

**Output SQL**

```
SQL: SELECT customers.customer_name FROM customers JOIN orders ON customers.
customer_id = orders.customer_id
WHERE orders.order_status = "On Road"
INTERSECT SELECT customers.customer_name FROM customers JOIN orders ON
customers.customer_id = orders.customer_id
WHERE orders.order_status = "Shipped"
```

Figure 8: An exemplar for text-to-SQL generation. The problem is taken from the Spider dataset (Yu et al., 2018). The problem description contains the database schema, and the model is required to predict the SQL query. The prompt includes the contents of one row from each table.

**C++ Program**

```cpp
string caesar_cipher ( string text,
int s ) {
  string result = "";
  for ( int i = 0;
  i < text . length ( );
  i ++ ) {
    if ( isupper ( text [ i ] ) )
    result += char ( int ( text [ i ]
    + s - 65 ) % 26 + 65 );
    else result += char ( int ( text [
     i ] + s - 97 ) % 26 + 97 );
  }
  return result;
}
```

**Python Program**

```python
def caesar_cipher(text, s):
    result = ''
    for i in range(len(text)):
        char = text[i]
        if char.isupper():
            result += chr(((((ord(char
            ) + s) - 65) % 26) + 65))
        else:
            result += chr(((((ord(char
            ) + s) - 97) % 26) + 97))
    return result
```

**Unit Tests**

```
assert caesar_cipher('35225904', 2) == 'ikhhkofj'
... (8 unit tests omitted)
assert caesar_cipher('11', 93) == 'tt'
```

Figure 9: An example from the TransCoder dataset. The problem description contains the C++ program and unit tests, and the model is required to predict the Python program.

**Original SQL**

Question: Give the names of countries with English and French as official languages.

```sql
SELECT country.name FROM country
JOIN countrylanguage ON country.code = countrylanguage.countrycode
WHERE countrylanguage.language = "English"
INTERSECT SELECT country.name FROM country
JOIN countrylanguage ON country.code = countrylanguage.countrycode
WHERE countrylanguage.language = "French"
```

**Code explanation**

The execution of the SQL query above would return a table with 1 column.
The first column, ''country.name'' would contain the country name.
With ''country JOIN countrylanguage'', the table would contain the data about countries and languages.
With ''WHERE countrylanguage.language = 'English''', the table filters the records to only include countries with the language ''English''.
With ''WHERE countrylanguage.language = 'French''', the table filters the records to only include countries with the language ''French''.
With ''INTERSECT'', the table would be filtered to only include countries with both ''English'' and ''French'' as languages.
So the SQL query returns a table with 1 column, the country name of countries with both ''English'' and ''French'' as languages.

**Self-debugging with explanation**

As in your explanation, the SQL query returns a table with 1 column, the country name of countries with both ''English'' and ''French'' as languages. The question returns the names of countries with English and French as official languages. So the SQL prediction above is wrong. Please fix the SQL.

- - - - - - - - - - - - - - - - - - - - - - - - - - - - - - - - - - - - - - - - - -

```sql
SELECT country.name FROM country JOIN countrylanguage ON country.code =
countrylanguage.countrycode
WHERE countrylanguage.language = "English"
AND countrylanguage.isofficial = "T"
INTERSECT SELECT country.name FROM country JOIN countrylanguage ON country.code
= countrylanguage.countrycode WHERE countrylanguage.language = "French"
AND countrylanguage.isofficial = "T"
```

Figure 10: An example on the Spider benchmark, where SELF-DEBUGGING fixes the prediction for an extra hard problem. Database information is omitted in the figure for clarity.

```
C++

int remainder_7_large_numbers ( string
 num ) {
  int series [ ] = {
    1, 3, 2, - 1, - 3, - 2 };
    int series_index = 0;
    int result = 0;
    for ( int i = num . size ( ) - 1;
    i >= 0; i -- ) {
      int digit = num [ i ] - '0';
      result += digit * series [
      series_index ];
      series_index = ( series_index +
      1 ) % 6;
      result %= 7;
    }
    if ( result < 0 ) result = (
    result + 7 ) % 7;
    return result;
  }
```

```
Original Python

def remainder_7_large_numbers(num):
    series = [1, 3, 2, -1, -3, -2]
    series_index = 0
    result = 0
    for i in range((len(num) - 1), -1,
     -1):
        digit = (num[i] - '0')
        result += (digit * series[
        series_index])
        series_index = ((series_index
        + 1) % 6)
        result %= 7
    if (result < 0):
        result = ((result + 7) % 7)
    return result
```

**Self-debugging with UT feedback**

```
The Python translation does not do the same thing as the C++ code. These are
the results of one failed unit test that tests whether the Python translation's
 outputs match the C++ program's outputs:
Failed: assert remainder_7_large_numbers('K') == 6
Actual Result: Python runtime error: Traceback (most recent call last):
  File <filename>, line 29, in <module>
    print(remainder_7_large_numbers('K') )
  File <filename>, line 22, in remainder_7_large_numbers
    digit = (num[i] - '0')
TypeError: unsupported operand type(s) for -: 'str' and 'str'
Correct the Python translation.
```

- - - - - - - - - - - - - - - - - - - - - - - - - - - - - - - - - - - - - - - - - - - - - - - - - - -

```
def remainder_7_large_numbers(num):
    series = [1, 3, 2, -1, -3, -2]
    series_index = 0
    result = 0
    for i in range((len(num) - 1), -1, -1):
        digit = (ord(num[i]) - ord('0'))
        result += (digit * series[series_index])
        series_index = ((series_index + 1) % 6)
        result %= 7
    if (result < 0):
        result = ((result + 7) % 7)
    return result
```

Figure 11: An example where SELF-DEBUGGING with unit test feedback fixes the code translation error, while the simple feedback fails.

C++

```
void function_copy_string ( char s1 [
], char s2 [ ], int index = 0 ) {
  s2 [ index ] = s1 [ index ];
  if ( s1 [ index ] == '\0' ) return;
  function_copy_string ( s1, s2, index
   + 1 );
}
```

Baseline pass@25

```
def function_copy_string(s1, s2, index
=0):
    s2[index] = s1[index]
    if (s1[index] == '\0'):
        return None
    function_copy_string(s1, s2, (
    index + 1))
```

+ Expl.

```
The code is an implementation of iterative function of copying a given string.
The character at the given index in the first string will be copied to the same
 index of the second string.
If the character at the given index in the first string is '\0', which means
the end of the string, the function will be returned.
Otherwise, the next character of the first string will be copied to the second
string.
```

```
def function_copy_string(s1, s2, idx=0):
    s2[idx] = s1[idx]
    if ((s1[idx] == '\0') or ((len(s1) - 1) == idx)):
        return
    function_copy_string(s1, s2, (idx + 1))
```

Figure 12: An example where adding code explanation results in the correct code translation, while the baseline prompt does not. Specifically, the translation with code explanation correctly checks for the end of a string.

Original Python code

```
assert encode_list
([1,1,2,3,4,4.3,5,1])==[[2, 1], [1,
2], [1, 3], [1, 4], [1, 4.3], [1, 5],
[1, 1]

Write a function to reflect the run-
length encoding from a list.
```
```
def encode_list(nums):
    res = []
    for i in nums:
        if i not in res:
            res.append([nums.count(i),
            i])
    return res
```

Prediction after self-debugging

```
def encode_list(nums):
  res = []
  count = 1
  for i in range(1, len(nums)):
    if nums[i] == nums[i-1]:
      count += 1
    else:
      res.append([count, nums[i-1]])
      count = 1
  res.append([count, nums[-1]])
  return res
```

Figure 13: An example on MBPP where the prediction after SELF-DEBUGGING is very different from the initial code.

# H TEXT-TO-SQL GENERATION

## H.1 BASELINE PROMPT (5-SHOT)

```
CREATE TABLE department (
department_id number ,
name text ,
creation text ,
ranking number ,
budget_in_billions number ,
num_employees number ,
primary key ( department_id )
)
insert into department (department_id, name, creation, ranking,
budget_in_billions, num_employees) values (1,'State','1789','1'
,9.9600000000000008526,30265.999999999999) ;

CREATE TABLE head (
head_id number ,
name text ,
born_state text ,
age number ,
primary key ( head_id )
)
insert into head (head_id, name, born_state, age) values (1,'Tiger Woods','
Alabama',66.999999999999999998) ;

CREATE TABLE management (
department_id number ,
head_id number ,
temporary_acting text ,
primary key ( department_id ) ,
foreign key ( head_id ) references head ( head_id )  ,
foreign key ( department_id ) references department ( department_id )
)
insert into management (department_id, head_id, temporary_acting) values
(2,5,'Yes') ;

Translate the following question into SQL.

Question: In which year were most departments established?

SQL: SELECT creation FROM department GROUP BY creation ORDER BY COUNT(*)
DESC LIMIT 1

CREATE TABLE city (
city_id number ,
official_name text ,
status text ,
area_km_2 number ,
population number ,
census_ranking text ,
primary key ( city_id )
)
insert into city (city_id, official_name, status, area_km_2, population,
census_ranking) values (1,'Grand Falls/Grand-Sault','Town','18.06','5706','
636 of 5008') ;

CREATE TABLE farm (
farm_id number ,
year number ,
```

```
total_horses number ,
working_horses number ,
total_cattle number ,
oxen number ,
bulls number ,
cows number ,
pigs number ,
sheep_and_goats number ,
primary key ( farm_id )
)
insert into farm (farm_id, year, total_horses, working_horses, total_cattle,
 oxen, bulls, cows, pigs, sheep_and_goats) values (1,'1927','5056.5','3900.1
','8374.5','805.5','31.6','3852.1','4412.4','7956.3') ;

CREATE TABLE farm_competition (
competition_id number ,
year number ,
theme text ,
host_city_id number ,
hosts text ,
primary key ( competition_id ) ,
foreign key ( host_city_id ) references city ( city_id )
)
insert into farm_competition (competition_id, year, theme, host_city_id,
hosts) values (1,'2013','Carnival M is back!',1,'Miley Cyrus Jared Leto and
Karen Mok') ;

CREATE TABLE competition_record (
competition_id number ,
farm_id number ,
rank number ,
primary key ( competition_id ) ,
foreign key ( farm_id ) references farm ( farm_id )  ,
foreign key ( competition_id ) references farm_competition ( competition_id
)
)
insert into competition_record (competition_id, farm_id, rank) values
(1,8,1) ;

Translate the following question into SQL.

Question: Show the status of the city that has hosted the greatest number of
 competitions.

SQL: SELECT city.status FROM city JOIN farm_competition ON city.city_id =
farm_competition.host_city_id GROUP BY farm_competition.host_city_id ORDER
BY COUNT(*) DESC LIMIT 1

CREATE TABLE customers (
customer_id number ,
customer_name text ,
customer_details text ,
primary key ( customer_id )
)
insert into customers (customer_id, customer_name, customer_details) values
(1, 'Savannah', 'rerum') ;

CREATE TABLE invoices (
invoice_number number ,
invoice_date time ,
invoice_details text ,
primary key ( invoice_number )
```

```
)
insert into invoices (invoice_number, invoice_date, invoice_details) values
(1, '1989-09-03 16:03:05', 'vitae') ;

CREATE TABLE orders (
order_id number ,
customer_id number ,
order_status text ,
date_order_placed time ,
order_details text ,
primary key ( order_id ),
foreign key ( customer_id ) references customers ( customer_id )
)
insert into orders (order_id, customer_id, order_status, date_order_placed,
order_details) values (1, 2, 'Shipped', '2009-02-21 15:26:19', NULL);

CREATE TABLE products (
product_id number ,
product_name text ,
product_details text,
primary key ( product_id )
)
insert into products (product_id, product_name, product_details) values (1,
'food', NULL);

CREATE TABLE order_items (
order_item_id number ,
product_id number ,
order_id number ,
order_item_status text ,
order_item_details text ,
primary key ( order_item_id ) ,
foreign key ( product_id ) references product ( product_id ) ,
foreign key ( order_id ) references orders ( order_id )
)
insert into order_items (order_item_id, product_id, order_id,
order_item_status, order_item_details) values (1, 4, 6, 'Finish', NULL) ;

CREATE TABLE shipments (
shipment_id number ,
order_id number ,
invoice_number number ,
shipment_tracking_number text ,
shipment_date time ,
other_shipment_details text ,
primary key ( shipment_id ) ,
foreign key ( invoice_number ) references invoices ( invoices_number ) ,
foreign key ( order_id ) references order ( order_id )
)
insert into shipments (shipment_id, order_id, invoice_number,
shipment_tracking_number, shipment_date, other_shipment_details) values (1,
5, 13, '3452', '1983-08-13 22:34:11', NULL) ;

CREATE TABLE shipment_items (
shipment_id number ,
order_item_id number ,
primary key ( ) ,
foreign key ( shipment_id ) references shipments ( shipment_id ) ,
foreign key ( order_item_id ) references order_items ( order_item_id )
)
insert into shipment_items (shipment_id, order_item_id) values (11, 12) ;

Translate the following question into SQL.
```

Question: Which customers have both "On Road" and "Shipped" as order status? List the customer names.

SQL: SELECT customers.customer_name FROM customers JOIN orders ON customers.customer_id = orders.customer_id WHERE orders.order_status = "On Road" INTERSECT SELECT customers.customer_name FROM customers JOIN orders ON customers.customer_id = orders.customer_id WHERE orders.order_status = "Shipped"

```sql
CREATE TABLE station (
id number ,
name text ,
lat number ,
long number ,
dock_count number ,
city text ,
installation_date text ,
primary key ( id )
)
insert into station (id, name, lat, long, dock_count, city,
installation_date) values (2,'San Jose Diridon Caltrain Station'
,37.329732,-1.21901782000000011405e+02,27,'San Jose','8/6/2013');

CREATE TABLE status (
station_id number ,
bikes_available number ,
docks_available number ,
time text ,
primary key ( ) ,
foreign key ( station_id ) references station ( id )
)
insert into status (station_id, bikes_available, docks_available,time values
 (3,12,3,'2015-06-02 12:46:02');

CREATE TABLE trip (
id number ,
duration number ,
start_date text ,
start_station_name text ,
start_station_id number ,
end_date text ,
end_station_name text ,
end_station_id number ,
bike_id number ,
subscription_type text ,
zip_code number ,
primary key ( id )
)
insert into trip (id, duration, start_date, start_station_name,
start_station_id, end_date, end_station_name, end_station_id, bike_id,
subscription_type, zip_code) values (900504,384,'8/21/2015 17:03','Howard at
 2nd',63,'8/21/2015 17:10','San Francisco Caltrain 2 (330 Townsend)',69,454,
'Subscriber',94041);
```

Translate the following question into SQL.

Question: What is the average longitude of stations that never had bike availability more than 10?

SQL: SELECT AVG(long) FROM station WHERE id NOT IN (SELECT station_id FROM status GROUP BY station_id HAVING MAX(bikes_available) > 10)

```
CREATE TABLE allergy_type (
allergy text ,
allergytype text ,
primary key ( allergy )
)
insert into allergy_type (allergy, allergytype) values ( 'Eggs', 'food' );

CREATE TABLE has_allergy (
stuid number ,
allergy text ,
foreign key ( allergy ) references allergy_type ( allergy ) ,
foreign key ( stuid ) references student ( stuid )
)
insert into has_allergy (stuid, allergy) values ( 1001, 'Cat' );

CREATE TABLE student (
stuid number ,
lname text ,
fname text ,
age number ,
sex text ,
major number ,
advisor number ,
city_code text ,
primary key ( stuid )
)
insert into student (stuid, lname, fname, age, sex, major, advisor,
city_code) values ( 1001, 'Smith', 'Linda', 18, 'F', 600, 1121,'BAL');

Translate the following question into SQL.

Question: Find the number of students who are older than 18 and do not have
allergy to either food or animal.

SQL: SELECT COUNT(*) FROM student WHERE age > 18 AND stuid NOT IN (SELECT
has_allergy.stuid FROM has_allergy JOIN allergy_type ON has_allergy.allergy
 = allergy_type.allergy WHERE allergy_type.allergytype = "food" OR
allergy_type.allergytype = "animal")

<insert database schemas and the new question here>
```

## H.2  SIMPLE FEEDBACK PROMPT (9-SHOT)

```
CREATE TABLE department (
department_id number ,
name text ,
creation text ,
ranking number ,
budget_in_billions number ,
num_employees number ,
primary key ( department_id )
)
insert into department (department_id, name, creation, ranking,
budget_in_billions, num_employees) values (1,'State','1789','1'
,9.9600000000000008526,30265.999999999999) ;

CREATE TABLE head (
```

```
head_id number ,
name text ,
born_state text ,
age number ,
primary key ( head_id )
)
insert into head (head_id, name, born_state, age) values (1,'Tiger Woods','
Alabama',66.999999999999999998) ;

CREATE TABLE management (
department_id number ,
head_id number ,
temporary_acting text ,
primary key ( department_id ) ,
foreign key ( head_id ) references head ( head_id )  ,
foreign key ( department_id ) references department ( department_id )
)
insert into management (department_id, head_id, temporary_acting) values
(2,5,'Yes') ;

Translate the following question into SQL.

Question: In which year were most departments established?

SQL: SELECT creation, COUNT(*) FROM department GROUP BY creation ORDER BY
COUNT(*) DESC LIMIT 1

Feedback: The SQL prediction above is wrong. Please fix the SQL.

SQL: SELECT creation FROM department GROUP BY creation ORDER BY COUNT(*)
DESC LIMIT 1

Feedback: The SQL prediction above is correct!

CREATE TABLE people (
people_id number ,
sex text ,
name text ,
height number ,
weight number ,
primary key ( people_id )
)
insert into people (people_id, sex, name, height, weight) values (1, M,
Hubert Henno, 188, 83);

Translate the following question into SQL.

Question: Find the average and minimum weight for each gender.

SQL: SELECT AVG(weight), MIN(weight), sex FROM people GROUP BY sex

Feedback: The SQL prediction above is correct!

CREATE TABLE customers (
customer_id number ,
customer_name text ,
customer_details text ,
primary key ( customer_id )
)
insert into customers (customer_id, customer_name, customer_details) values
(1, 'Savannah', 'rerum') ;
```

```
CREATE TABLE invoices (
invoice_number number ,
invoice_date time ,
invoice_details text ,
primary key ( invoice_number )
)
insert into invoices (invoice_number, invoice_date, invoice_details) values
(1, '1989-09-03 16:03:05', 'vitae') ;

CREATE TABLE orders (
order_id number ,
customer_id number ,
order_status text ,
date_order_placed time ,
order_details text ,
primary key ( order_id ),
foreign key ( customer_id ) references customers ( customer_id )
)
insert into orders (order_id, customer_id, order_status, date_order_placed,
order_details) values (1, 2, 'Shipped', '2009-02-21 15:26:19', NULL);

CREATE TABLE products (
product_id number ,
product_name text ,
product_details text,
primary key ( product_id )
)
insert into products (product_id, product_name, product_details) values (1,
'food', NULL);

CREATE TABLE order_items (
order_item_id number ,
product_id number ,
order_id number ,
order_item_status text ,
order_item_details text ,
primary key ( order_item_id ) ,
foreign key ( product_id ) references product ( product_id ) ,
foreign key ( order_id ) references orders ( order_id )
)
insert into order_items (order_item_id, product_id, order_id,
order_item_status, order_item_details) values (1, 4, 6, 'Finish', NULL) ;

CREATE TABLE shipments (
shipment_id number ,
order_id number ,
invoice_number number ,
shipment_tracking_number text ,
shipment_date time ,
other_shipment_details text ,
primary key ( shipment_id ) ,
foreign key ( invoice_number ) references invoices ( invoices_number ) ,
foreign key ( order_id ) references order ( order_id )
)
insert into shipments (shipment_id, order_id, invoice_number,
shipment_tracking_number, shipment_date, other_shipment_details) values (1,
5, 13, '3452', '1983-08-13 22:34:11', NULL) ;

CREATE TABLE shipment_items (
shipment_id number ,
order_item_id number ,
primary key ( ) ,
foreign key ( shipment_id ) references shipments ( shipment_id ) ,
```

```
foreign key ( order_item_id ) references order_items ( order_item_id )
)
insert into shipment_items (shipment_id, order_item_id) values (11, 12) ;

Translate the following question into SQL.

Question: Which customers have both "On Road" and "Shipped" as order status?
 List the customer names.

SQL: SELECT customers.customer_name FROM customers JOIN orders ON customers.
customer_id = orders.customer_id WHERE orders.order_status = "On Road" AND
orders.order_status = "Shipped"

Feedback: The SQL prediction above is wrong. Please fix the SQL.

SQL: SELECT customers.customer_name FROM customers JOIN orders ON customers.
customer_id = orders.customer_id WHERE orders.order_status = "On Road"
INTERSECT SELECT customers.customer_name FROM customers JOIN orders ON
customers.customer_id = orders.customer_id WHERE orders.order_status = "
Shipped"

Feedback: The SQL prediction above is correct!

CREATE TABLE flight (
flno number ,
origin text ,
destination text ,
aid number ,
primary key ( flno )
)
insert into flight (flno, origin, destination, aid) values (99.0, Los
Angeles, Washington D.C., 1);

Translate the following question into SQL.

Question: Show origins of all flights with destination Honolulu.

SQL: SELECT origin FROM flight WHERE destination = "HONO"

Feedback: The SQL prediction above is wrong. Please fix the SQL.

SQL: SELECT origin FROM flight WHERE destination = "Honolulu"

Feedback: The SQL prediction above is correct!

CREATE TABLE station (
id number ,
name text ,
lat number ,
long number ,
dock_count number ,
city text ,
installation_date text ,
primary key ( id )
)
insert into station (id, name, lat, long, dock_count, city,
installation_date) values (2, San Jose Diridon Caltrain Station,
37.329732,-1.21901782000000011405e+02,27, San Jose, 8/6/2013);

CREATE TABLE status (
station_id number ,
```

```
bikes_available number ,
docks_available number ,
time text ,
primary key ( ) ,
foreign key ( station_id ) references station ( id )
)
insert into status (station_id, bikes_available, docks_available,time values
 (3,12,3, 2015-06-02 12:46:02);

Translate the following question into SQL.

Question: What is the average longitude of stations that never had bike
availability more than 10?

SQL: SELECT AVG(long) FROM station WHERE id IN (SELECT station_id FROM
status WHERE bikes_available <= 10)

Feedback: The SQL prediction above is wrong. Please fix the SQL.

SQL: SELECT AVG(long) FROM station WHERE id NOT IN (SELECT station_id FROM
status WHERE bikes_available > 10)

Feedback: The SQL prediction above is correct!

CREATE TABLE host (
host_id number ,
name text ,
nationality text ,
age number ,
primary key ( host_id )
)
insert into host (host_id, name, nationality, age) values (1,"Austin Daye","
United States",43);

Translate the following question into SQL.

Question: Show the name and the nationality of the oldest host.

SQL: SELECT name, nationality FROM host WHERE age = (SELECT MIN(age) FROM
host)

Feedback: The SQL prediction above is wrong. Please fix the SQL.

SQL: SELECT name, nationality FROM host ORDER BY age DESC LIMIT 1

Feedback: The SQL prediction above is correct!

CREATE TABLE city (
city_id number ,
official_name text ,
status text ,
area_km_2 number ,
population number ,
census_ranking text ,
primary key ( city_id )
)
insert into city (city_id, official_name, status, area_km_2, population,
census_ranking) values (1,'Grand Falls/Grand-Sault','Town','18.06','5706','
636 of 5008') ;

CREATE TABLE farm (
```

```
farm_id number ,
year number ,
total_horses number ,
working_horses number ,
total_cattle number ,
oxen number ,
bulls number ,
cows number ,
pigs number ,
sheep_and_goats number ,
primary key ( farm_id )
)
insert into farm (farm_id, year, total_horses, working_horses, total_cattle,
 oxen, bulls, cows, pigs, sheep_and_goats) values (1,'1927','5056.5','3900.1
','8374.5','805.5','31.6','3852.1','4412.4','7956.3') ;

CREATE TABLE farm_competition (
competition_id number ,
year number ,
theme text ,
host_city_id number ,
hosts text ,
primary key ( competition_id ) ,
foreign key ( host_city_id ) references city ( city_id )
)
insert into farm_competition (competition_id, year, theme, host_city_id,
hosts) values (1,'2013','Carnival M is back!',1,'Miley Cyrus Jared Leto and
Karen Mok') ;

CREATE TABLE competition_record (
competition_id number ,
farm_id number ,
rank number ,
primary key ( competition_id ) ,
foreign key ( farm_id ) references farm ( farm_id )  ,
foreign key ( competition_id ) references farm_competition ( competition_id
)
)
insert into competition_record (competition_id, farm_id, rank) values
(1,8,1) ;

Translate the following question into SQL.

Question: How many different statuses do cities have?

SQL: SELECT COUNT(status) FROM city

Feedback: The SQL prediction above is wrong. Please fix the SQL.

SQL: SELECT COUNT(DISTINCT status) FROM city

Feedback: The SQL prediction above is correct!

CREATE TABLE user_profiles (
uid number ,
name text ,
followers number ,
primary key ( uid )
)

CREATE TABLE tweets (
id number ,
```

```
uid number ,
primary key ( id ) ,
foreign key ( uid ) references user_profiles ( uid )
)

Translate the following question into SQL.

Question: Find the average number of followers for the users who had some
tweets.

SQL: SELECT AVG(followers) FROM user_profiles

Feedback: The SQL prediction above is wrong. Please fix the SQL.

SQL: SELECT AVG(followers) FROM user_profiles WHERE uid IN (SELECT uid FROM
tweets)

Feedback: The SQL prediction above is correct!

CREATE TABLE trip (
id number ,
duration number ,
bike_id number ,
primary key ( id )
)
insert into trip (id, duration, bike_id) values (900504, 384, 454);

Translate the following question into SQL.

Question: What is the total and maximum duration for all trips with the bike
 id 636?

SQL: SELECT SUM(duration), MAX(duration) FROM trip WHERE bike_id = 636

Feedback: The SQL prediction above is correct!

<insert database schemas and the new question here>
<insert original SQL here>
```

## H.3 EXPLANATION FEEDBACK PROMPT (9-SHOT)

```
CREATE TABLE department (
department_id number ,
name text ,
creation text ,
ranking number ,
budget_in_billions number ,
num_employees number ,
primary key ( department_id )
)
insert into department (department_id, name, creation, ranking,
budget_in_billions, num_employees) values (1,'State','1789','1'
,9.9600000000000008526,30265.999999999999999) ;

CREATE TABLE head (
head_id number ,
name text ,
```

```
born_state text ,
age number ,
primary key ( head_id )
)
insert into head (head_id, name, born_state, age) values (1,'Tiger Woods','
Alabama',66.999999999999999998) ;

CREATE TABLE management (
department_id number ,
head_id number ,
temporary_acting text ,
primary key ( department_id ) ,
foreign key ( head_id ) references head ( head_id )  ,
foreign key ( department_id ) references department ( department_id )
)
insert into management (department_id, head_id, temporary_acting) values
(2,5,'Yes') ;
```

Translate the following question into SQL.

Question: In which year were most departments established?

SQL: SELECT creation, COUNT(*) FROM department GROUP BY creation ORDER BY
COUNT(*) DESC LIMIT 1

The execution of the SQL query above would return a table with 2 columns.
The first column, "creation" would contain the year in which a department
was created. The second column, "COUNT(*)" would contain the number of
departments created in that year. With "GROUP BY creation ORDER BY COUNT(*)
DESC", the table would be sorted in descending order by the number of
departments created in each year. With "LIMIT 1", only the first (i.e.,
highest) result would be returned. So the SQL query returns a table with 2
columns, the year in which most departments were created, and the number of
departments created in that year.

Feedback: As in your explanation, the SQL query returns a table with 2
columns, the year in which most departments was created, and the number of
departments created in that year. The question returns 1 column, the year in
 which the most departments were established. So the SQL prediction above is
 wrong. Please fix the SQL.

SQL: SELECT creation FROM department GROUP BY creation ORDER BY COUNT(*)
DESC LIMIT 1

The execution of the SQL query above would return a table with 1 column. The
 first column, "creation" would contain the year in which a department was
created. With "GROUP BY creation ORDER BY COUNT(*) DESC", the table would be
 sorted in descending order by the number of departments created in each
year. With "LIMIT 1", only the first (i.e., highest) result would be
returned. So the SQL query returns a table with 1 column, the year in which
most departments were created.

Feedback: As in your explanation, the SQL query returns a table with 1
column, the year in which most departments was created. The question returns
 1 column, the year in which the most departments were established. So the
SQL prediction above is correct!

CREATE TABLE people (
people_id number ,
sex text ,
name text ,
height number ,
```

```
weight number ,
primary key ( people_id )
)
insert into people (people_id, sex, name, height, weight) values (1, M,
Hubert Henno, 188, 83);

Translate the following question into SQL.

Question: Find the average and minimum weight for each gender.

SQL: SELECT AVG(weight), MIN(weight), sex FROM people GROUP BY sex

The execution of the SQL query above would return a table with 3 columns.
The first column, "AVG(weight)" would contain the average weight. The second
 column, "MIN(weight)" would contain the minimum weight. The third column, "
sex" would contain the gender. With "GROUP BY sex", the table would be
grouped by gender. So the SQL query returns a table with 3 columns, the
average weight, the minimum weight, and the gender of people for each gender
.

Feedback: As in your explanation, the SQL query returns a table with 3
columns, the average weight, the minimum weight, and the gender of people
for each gender. The question returns 3 columns, the average and minimum
weight for each gender. So the SQL prediction above is correct!

CREATE TABLE customers (
customer_id number ,
customer_name text ,
customer_details text ,
primary key ( customer_id )
)
insert into customers (customer_id, customer_name, customer_details) values
(1, 'Savannah', 'rerum') ;

CREATE TABLE invoices (
invoice_number number ,
invoice_date time ,
invoice_details text ,
primary key ( invoice_number )
)
insert into invoices (invoice_number, invoice_date, invoice_details) values
(1, '1989-09-03 16:03:05', 'vitae') ;

CREATE TABLE orders (
order_id number ,
customer_id number ,
order_status text ,
date_order_placed time ,
order_details text ,
primary key ( order_id ),
foreign key ( customer_id ) references customers ( customer_id )
)
insert into orders (order_id, customer_id, order_status, date_order_placed,
order_details) values (1, 2, 'Shipped', '2009-02-21 15:26:19', NULL);

CREATE TABLE products (
product_id number ,
product_name text ,
product_details text,
primary key ( product_id )
)
insert into products (product_id, product_name, product_details) values (1,
'food', NULL);
```

```
CREATE TABLE order_items (
order_item_id number ,
product_id number ,
order_id number ,
order_item_status text ,
order_item_details text ,
primary key ( order_item_id ) ,
foreign key ( product_id ) references product ( product_id ) ,
foreign key ( order_id ) references orders ( order_id )
)
insert into order_items (order_item_id, product_id, order_id,
order_item_status, order_item_details) values (1, 4, 6, 'Finish', NULL) ;

CREATE TABLE shipments (
shipment_id number ,
order_id number ,
invoice_number number ,
shipment_tracking_number text ,
shipment_date time ,
other_shipment_details text ,
primary key ( shipment_id ) ,
foreign key ( invoice_number ) references invoices ( invoices_number ) ,
foreign key ( order_id ) references order ( order_id )
)
insert into shipments (shipment_id, order_id, invoice_number,
shipment_tracking_number, shipment_date, other_shipment_details) values (1,
5, 13, '3452', '1983-08-13 22:34:11', NULL) ;

CREATE TABLE shipment_items (
shipment_id number ,
order_item_id number ,
primary key ( ) ,
foreign key ( shipment_id ) references shipments ( shipment_id ) ,
foreign key ( order_item_id ) references order_items ( order_item_id )
)
insert into shipment_items (shipment_id, order_item_id) values (11, 12) ;

Translate the following question into SQL.

Question: Which customers have both "On Road" and "Shipped" as order status?
 List the customer names.

SQL: SELECT customers.customer_name FROM customers JOIN orders ON customers.
customer_id = orders.customer_id WHERE orders.order_status = "On Road" AND
orders.order_status = "Shipped"
```

The execution of the SQL query above would return an empty table. The first
column, "customers.customer_name" would contain the customer name. With "
customers JOIN orders", the table would contain the data about customers
with orders. In the WHERE clause, with "orders.order_status = 'On Road'",
the table filters the records to only include customers who have order
status "On Road". With "orders.order_status = 'Shipped'", the table filters
the records to only include customers who have order status "Shipped". With
"WHERE orders.order_status = 'On Road' AND orders.order_status = 'Shipped'",
 the table filters the records to only include customers who have both "On
Road" and "Shipped" as order status at the same time. This is not possible,
since the two are mutually exclusive. So the SQL query returns an empty
table, because the "WHERE" clause contains mutually exclusive conditions.

Feedback: As in your explanation, the SQL query returns an empty table,
because the "WHERE" clause contains mutually exclusive conditions. The
question returns 1 column, the customer names who have order status of both

"On Road" and "Shipped". So the question returns 1 column. So the SQL prediction above is wrong. Please fix the SQL.

**SQL:** SELECT customers.customer_name FROM customers JOIN orders ON customers.customer_id = orders.customer_id WHERE orders.order_status = "On Road" INTERSECT SELECT customers.customer_name FROM customers JOIN orders ON customers.customer_id = orders.customer_id WHERE orders.order_status = "Shipped"

The execution of the SQL query above would return a table with 1 column. The first column, "customers.customer_name" would contain the customer names. With "customers JOIN orders", the table would contain the data about customers with orders. With "WHERE orders.order_status = 'On Road'", the table filters the records to only include customers who have order status "On Road". With "WHERE orders.order_status = 'Shipped'", the table filters the records to only include customers who have order status "Shipped". With "INTERSECT", the table would be filtered to only include customers who have both "On Road" and "Shipped" as order status. So the SQL query returns a table with 1 column, the customer names who have the order status "On Road" and "Shipped".

**Feedback:** As in your explanation, the SQL query returns a table with 1 column, the customer names who have the order status "On Road" and "Shipped". The question returns 1 column, the customer names who have order status of both "On Road" and "Shipped". So the SQL prediction above is correct!

CREATE TABLE flight (
flno number ,
origin text ,
destination text ,
aid number ,
primary key ( flno )
)
insert into flight (flno, origin, destination, aid) values (99.0, Los Angeles, Washington D.C., 1);

Translate the following question into SQL.

**Question:** Show origins of all flights with destination Honolulu.

**SQL:** SELECT origin FROM flight WHERE destination = "HONO"

The execution of the SQL query above would return an empty table. The first column, "origin" would contain the origin of flights. With "WHERE destination = 'HONO'", the table filters the records to only include flights whose destination is "HONO". So the SQL query returns an empty table.

**Feedback:** As in your explanation, the SQL query returns an empty table. The question returns 1 column, the origins of all flights with destination Honolulu. So the SQL prediction above is wrong. Please fix the SQL.

**SQL:** SELECT origin FROM flight WHERE destination = "Honolulu"

The execution of the SQL query above would return a table with 1 column. The first column, "origin" would contain the origin of flights. With "WHERE destination = 'Honolulu'", the table filters the records to only include flights whose destination is "Honolulu". So the SQL query returns a table with 1 column, the origin of flights with destination Honolulu.

**Feedback:** As in your explanation, the SQL query returns a table with 1 column, the origin of flights with destination Honolulu. The question returns 1 column, the origins of all flights with destination Honolulu. So the SQL prediction above is correct!

```
CREATE TABLE station (
id number ,
name text ,
lat number ,
long number ,
dock_count number ,
city text ,
installation_date text ,
primary key ( id )
)
insert into station (id, name, lat, long, dock_count, city,
installation_date) values (2, San Jose Diridon Caltrain Station,
37.329732,-1.21901782000000011405e+02,27, San Jose, 8/6/2013);

CREATE TABLE status (
station_id number ,
bikes_available number ,
docks_available number ,
time text ,
primary key ( ) ,
foreign key ( station_id ) references station ( id )
)
insert into status (station_id, bikes_available, docks_available,time values
 (3,12,3, 2015-06-02 12:46:02);

Translate the following question into SQL.

Question: What is the average longitude of stations that never had bike
availability more than 10?

SQL: SELECT AVG(long) FROM station WHERE id IN (SELECT station_id FROM
status WHERE bikes_available <= 10)

The execution of the SQL query above would return a table with 1 column. The
 first column, "AVG(long)" would contain the average longitude. With "WHERE
id IN (SELECT station_id FROM status WHERE bikes_available <= 10)", the
table filters the records to only include stations with 10 or less bikes
available. So the SQL query returns a table with 1 column, the average
longitude of stations with 10 or less bikes available.

Feedback: As in your explanation, the SQL query returns a table with 1
column, the average longitude of stations with 10 or less bikes available.
The question returns 1 column, the average longitude of stations that never
had bike availability more than 10. So the SQL prediction above is wrong.
Please fix the SQL.

SQL: SELECT AVG(long) FROM station WHERE id NOT IN (SELECT station_id FROM
status WHERE bikes_available > 10)

The execution of the SQL query above would return a table with 1 column. The
 first column, "AVG(long)" would contain the average longitude. With "WHERE
id NOT IN (SELECT station_id FROM status WHERE bikes_available > 10)", the
table filters the records to only include stations that do not have more
than 10 bikes available. So the SQL query returns a table with 1 column, the
 average longitude of stations that do not have more than 10 bikes available
.

Feedback: As in your explanation, the SQL query returns a table with 1
column, the average longitude of stations that do not have more than 10
bikes available. The question returns 1 column, the average longitude of
stations that never had bike availability more than 10. The question returns
 1 column, the average longitude of stations that never had bike
availability more than 10. So the SQL prediction above is correct!
```

```
CREATE TABLE host (
host_id number ,
name text ,
nationality text ,
age number ,
primary key ( host_id )
)
insert into host (host_id, name, nationality, age) values (1,"Austin Daye","
United States",43);

Translate the following question into SQL.

Question: Show the name and the nationality of the oldest host.

SQL: SELECT name, nationality FROM host WHERE age = (SELECT MIN(age) FROM
host)

The execution of the SQL query above would return a table with 2 columns.
The first column, "name" would contain the name. The second column, "
nationality" would contain the nationality. With "WHERE age = (SELECT MIN(
age) FROM host)", the table filters the records to only include hosts with
the minimum age, which means the youngest age. So the SQL query returns a
table with 2 columns, the name and the nationality of the host with the
youngest age.

Feedback: As in your explanation, the SQL query returns a table with 2
columns, the name and the nationality of the host with the youngest age. The
 question returns 2 columns, the name and the nationality of the oldest host
. So the SQL prediction above is wrong. Please fix the SQL.

SQL: SELECT name, nationality FROM host ORDER BY age DESC LIMIT 1

The execution of the SQL query above would return a table with 2 columns.
The first column, "name" would contain the name. The second column, "
nationality" would contain the nationality. With "ORDER BY age DESC", the
table is sorted in descending order. With "LIMIT 1", the table only includes
 the first record, which is the oldest host. So the SQL query returns a
table with 2 columns, the name and the nationality of the oldest host.

Feedback: As in your explanation, the SQL query returns a table with 2
columns, the name and the nationality of the oldest host. The question
returns 2 columns, the name and the nationality of the oldest host. So the
SQL prediction above is correct!

CREATE TABLE city (
city_id number ,
official_name text ,
status text ,
area_km_2 number ,
population number ,
census_ranking text ,
primary key ( city_id )
)
insert into city (city_id, official_name, status, area_km_2, population,
census_ranking) values (1,'Grand Falls/Grand-Sault','Town','18.06','5706','
636 of 5008') ;

CREATE TABLE farm (
farm_id number ,
year number ,
total_horses number ,
working_horses number ,
```

```
total_cattle number ,
oxen number ,
bulls number ,
cows number ,
pigs number ,
sheep_and_goats number ,
primary key ( farm_id )
)
insert into farm (farm_id, year, total_horses, working_horses, total_cattle,
 oxen, bulls, cows, pigs, sheep_and_goats) values (1,'1927','5056.5','3900.1
','8374.5','805.5','31.6','3852.1','4412.4','7956.3') ;

CREATE TABLE farm_competition (
competition_id number ,
year number ,
theme text ,
host_city_id number ,
hosts text ,
primary key ( competition_id ) ,
foreign key ( host_city_id ) references city ( city_id )
)
insert into farm_competition (competition_id, year, theme, host_city_id,
hosts) values (1,'2013','Carnival M is back!',1,'Miley Cyrus Jared Leto and
Karen Mok') ;

CREATE TABLE competition_record (
competition_id number ,
farm_id number ,
rank number ,
primary key ( competition_id ) ,
foreign key ( farm_id ) references farm ( farm_id )  ,
foreign key ( competition_id ) references farm_competition ( competition_id
)
)
insert into competition_record (competition_id, farm_id, rank) values
(1,8,1) ;

Translate the following question into SQL.

Question: How many different statuses do cities have?

SQL: SELECT COUNT(status) FROM city

The execution of the SQL query above would return a table with 1 column. The
 first column, "COUNT(status)" would contain the number of cities. So the
SQL query returns a table with 1 column, the number of cities.

Feedback: As in your explanation, the SQL query returns a table with 1
column, the number of cities. The question returns 1 column, the number of
different statuses of cities. So the SQL prediction above is wrong. Please
fix the SQL.

SQL: SELECT COUNT(DISTINCT status) FROM city

The execution of the SQL query above would return a table with 1 column. The
 first column, "COUNT(DISTINCT status)" would contain the number of
different statuses of cities. So the SQL query returns a table with 1 column
, the number of different statuses of cities.

Feedback: As in your explanation, the SQL query returns a table with 1
column, the number of different statuses of cities. The question returns 1
column, the number of different statuses of cities. So the SQL prediction
above is correct!
```

```
CREATE TABLE user_profiles (
uid number ,
name text ,
followers number ,
primary key ( uid )
)

CREATE TABLE tweets (
id number ,
uid number ,
primary key ( id ) ,
foreign key ( uid ) references user_profiles ( uid )
)

Translate the following question into SQL.

Question: Find the average number of followers for the users who had some
tweets.

SQL: SELECT AVG(followers) FROM user_profiles

The execution of the SQL query above would return a table with 1 column. The
 first column, "AVG(followers)" would contain the average number of
followers. So the SQL query returns a table with 1 column, the average
number of followers.

Feedback: As in your explanation, the SQL query returns a table with 1
column, the average number of followers. The question returns 1 column, the
average number of followers for the users who had some tweets. So the SQL
prediction above is wrong. Please fix the SQL.

SQL: SELECT AVG(followers) FROM user_profiles WHERE uid IN (SELECT uid FROM
tweets)

The execution of the SQL query above would return a table with 1 column. The
 first column, "AVG(followers)" would contain the average number of
followers. With "WHERE uid IN (SELECT uid FROM tweets)", the table filters
the records to only include users who have tweets. So the SQL query returns
a table with 1 column, the average number of followers of users who have
tweets.

Feedback: As in your explanation, the SQL query returns a table with 1
column, the average number of followers of users who have tweets. The
question returns 1 column, the average number of followers for the users who
 had some tweets. So the SQL prediction above is correct!

CREATE TABLE trip (
id number ,
duration number ,
bike_id number ,
primary key ( id )
)
insert into trip (id, duration, bike_id) values (900504, 384, 454);

Translate the following question into SQL.

Question: What is the total and maximum duration for all trips with the bike
 id 636?

SQL: SELECT SUM(duration), MAX(duration) FROM trip WHERE bike_id = 636
```

```
The execution of the SQL query above would return a table with 2 columns.
The first column, "SUM(duration)" would contain the sum of the duration. The
 second column, "MAX(duration)" would contain the maximum duration. With "
WHERE bike_id = 636", the table filters the records to only include trips
with bike ID 636. So the SQL query returns a table with 2 columns, the sum
of the duration and the maximum duration of trips with bike ID 636.

Feedback: As in your explanation, the SQL query returns a table with 2
columns, the sum of the duration and the maximum duration of trips with bike
 ID 636. The question returns 2 columns, the total and maximum duration for
all trips with the bike id 636. So the SQL prediction above is correct!

<insert database schemas and the new question here>
<insert original SQL here>
```

## H.4 PROMPT FOR QUESTION EXPLANATION

```
Infer the return type of the question.

CREATE TABLE song (
song_name text ,
artist_name text ,
country text ,
f_id number ,
genre_is text ,
rating number ,
languages text ,
primary key ( f_id )
)
insert into song (song_name, artist_name, country, f_id, genre_is, rating,
languages) values (Tumi robe nirobe, Shrikanta, India, 1, tagore, 8, bangla)
;

Question: What is the id, genre, and name of the artist for every English
song ordered by ascending rating?
Answer: "What is the id" returns 1 column. "What is the genre" returns 1
column. "What is the id, genre" returns 2 columns. "What is the name"
returns 1 column. "What is the id, genre, and name" returns 3 columns.The
question returns the id, genre, and name of the artist for every English
song ordered by ascending rating. So the question returns 3 columns.

CREATE TABLE physician (
employee_id number ,
name text ,
position text ,
primary key ( employee_id )
)
insert into physician (employee_id, name, position) values (1, John Dorian,
Staff Internist);

CREATE TABLE procedures (
code number ,
name text ,
cost number ,
primary key ( code )
)
insert into procedures (code, name, cost) values (1, Reverse Rhinopodoplasty
, 1500.0);
```

```
CREATE TABLE trained_in (
physician number ,
treatment text ,
primary key ( physician ) ,
foreign key ( physician ) references physician ( employee_id )
)
insert into trained_in (physician, treatment) values (3,1);
```

Question: Among the procedures that cost more than 1000, which were not specialized in by physician John Wen?
Answer: "Among the procedures that cost more than 1000, which were not specialized in by physician John Wen" returns 1 column. The question returns the procedures that cost more than 1000, and were not specialized in by physician John Wen. So the question returns 1 column.

```
CREATE TABLE department (
department_id number ,
name text ,
creation text ,
ranking number ,
budget_in_billions number ,
num_employees number ,
primary key ( department_id )
)
insert into department (department_id, name, creation, ranking,
budget_in_billions, num_employees) values (1, State, 1789, 1,
9.9600000000000008526, 30265.999999999999999) ;

CREATE TABLE head (
head_id number ,
name text ,
born_state text ,
age number ,
primary key ( head_id )
)
insert into head (head_id, name, born_state, age) values (1, Tiger Woods,
Alabama, 66.999999999999999998) ;

CREATE TABLE management (
department_id number ,
head_id number ,
temporary_acting text ,
primary key ( department_id ) ,
foreign key ( head_id ) references head ( head_id )  ,
foreign key ( department_id ) references department ( department_id )
)
insert into management (department_id, head_id, temporary_acting) values
(2,5, Yes) ;
```

Question: Which department has more than 1 head at a time? List the id, name and the number of heads.
Answer: "List the id" returns 1 column. "List the name" returns 1 column. "List the id, name" returns 2 columns. "List the number of heads" returns 1 column. "List the id, name and the number of heads" returns 3 columns. The question returns the id, name and the number of heads of the department has more than 1 head at a time. So the question returns 3 columns.

Question: Show the name and number of employees for the departments managed by heads whose temporary acting value is 'Yes'?
Answer: "Show the name" returns 1 column. "Show the number of employees" returns 1 column. "Show the name and number of employees" returns 2 columns.

The question returns the name and number of employees for the departments managed by heads whose temporary acting value is 'Yes'. So the question returns 2 columns.

```
CREATE TABLE class (
class_code text ,
crs_code text ,
prof_num number ,
primary key ( class_code ) ,
foreign key ( prof_num ) references professor ( emp_num ) ,
foreign key ( crs_code ) references course ( crs_code )
)
insert into class (class_code, crs_code, prof_num) values (10012, ACCT-211,
105);

CREATE TABLE employee (
emp_num number ,
emp_lname text ,
emp_fname text ,
emp_initial text ,
emp_jobcode text ,
primary key ( emp_num )
)
insert into employee (emp_num, emp_lname, emp_fname, emp_initial,
emp_jobcode values (100, Worley, James, F, CUST);

CREATE TABLE course (
crs_code text ,
dept_code text ,
crs_description text ,
primary key ( crs_code )
)
insert into course ( crs_code, dept_code, crs_description) values (ACCT-211,
 ACCT, Accounting I);

CREATE TABLE professor (
emp_num number ,
dept_code text ,
prof_office text ,
primary key ( emp_num ) ,
foreign key ( emp_num ) references employee ( emp_num )
)
insert into professor (emp_num, dept_code, prof_office ) values (103, HIST,
DRE 156);
```

Question: Find the first names and offices of all instructors who have taught some course and also find the course description.
Answer: "Find the first names" returns 1 column. "Find the offices" returns 1 column. "Find the first names and offices" returns 2 columns. "Find the first names and offices of all instructors who have taught some course" returns 2 columns. "and also find the course description" returns 1 column. "Find the first names and offices of all instructors who have taught some course and also find the course description" returns 3 columns. The question returns the first names and offices of instructors who have taught some course and the course description. So the question returns 3 columns.

```
CREATE TABLE airports (
apid number ,
name text ,
city text ,
country text ,
```

```
elevation text ,
primary key ( apid )
)

Question: List the id, country, city and name of the airports ordered
alphabetically by the name.
Answer: "List the id" returns 1 column. "List the country" returns 1 column.
 "List the id, country" returns 2 columns. "List the city" returns 1 column.
 "List the id, country, city" returns 3 columns. "List the name" returns 1
column. "List the id, country, city, and name" returns 4 columns. The
question returns the id, country, city and name of the airports ordered
alphabetically by the name. So the question returns 4 columns.

CREATE TABLE student (
stuid number ,
lname text ,
fname text ,
primary key ( stuid )
)
insert into student (stuid, lname, fname) values (1001, Smith, Linda);

CREATE TABLE plays_games (
stuid number ,
gameid number ,
hours_played number ,
population number ,
primary key ( stuid ) ,
foreign key ( stuid ) references student ( stuid )
)
insert into plays_games (stuid, gameid, hours_played) values (1001, 1, 35);

CREATE TABLE sports_info (
stuid number ,
sport_name text ,
hours_per_week number ,
games_played number ,
primary key ( stuid ) ,
foreign key ( stuid ) references student ( stuid )
)
insert into sport_info (stuid, sport_name, hours_per_week, games_played)
values (1001, Athletics, 2, 5);

Question: What are the ids of all students and number of hours played?
Answer: "What are the ids" returns 1 column. "What are the number of hours
played" returns 1 column. "What are the ids and number of hours played"
returns 2 columns. The question returns the ids of students and number of
hours played. So the question returns 2 columns.

Question: What is the total number of hours per week and number of games
played by students under 20?
Answer: "What is the total number of hours per week" returns 1 column. "What
 is the number of games played" returns 1 column. "What is the total number
of hours per week and number of games played" returns 2 columns. The
question returns the total number of hours per week and number of games
played by students under 20. So the question returns 2 columns.

CREATE TABLE journal (
journal_id number ,
theme text ,
primary id ( journal_id )
)
```

```
insert into journal (journal_id, theme) values (1, Miami Dolphins);

CREATE TABLE editor (
editor_id number ,
name text ,
age number ,
primary id ( editor_id )
)
insert into editor (editor_id, name, age) values (1.0, Kamila Porczyk, 34.0)
;

CREATE TABLE journal_committee (
editor_id number ,
journal_id number ,
primary id ( editor_id ) ,
foreign key ( editor_id ) references editor ( editor_id ) ,
foreign key ( journal_id ) references journal ( journal_id )
)
insert into journal_committee (editor_id, journal_id) values (1, 13);

Question: Show the names and ages of editors and the theme of journals for
which they serve on committees, in ascending alphabetical order of theme.
Answer: "Show the names" returns 1 column. "Show the ages" returns 1 column.
 "Show the names and ages" returns 2 columns. "Show the theme of journals"
returns 1 column. "Show the names and ages of editors and the theme of
journals" returns 3 columns. "Show the names and ages of editors and the
theme of journals for which they serve on committees" returns 3 columns. "
Show the names and ages of editors and the theme of journals for which they
serve on committees, in ascending alphabetical order of theme" returns 3
columns. The question returns the names and ages of editors and the theme of
 journals for which they serve on committees, in ascending alphabetical
order of theme. So the question returns 3 columns.

CREATE TABLE people (
people_id number ,
sex text ,
name text ,
height number ,
weight number ,
primary key ( people_id )
)
insert into people (people_id, sex, name, height, weight) values (1, M,
Hubert Henno, 188, 83);

Question: Find the average and minimum weight for each gender.
Answer: "Find the average weight" returns 1 column. "Find the minimum weight
" returns 1 column. "Find the average and minimum weight" returns 2 columns.
 "Find the average and minimum weight for each gender" returns 3 columns.
The question returns the average and minimum weight for each gender. So the
question returns 3 columns.

Question: List all info about all people.
Answer: "List all info" returns all columns. The question returns all info
about all people. So the question returns all columns.

CREATE TABLE student (
stuid number ,
lname text ,
fname text ,
age number ,
sex text ,
```

```
major number ,
advisor number ,
city_code text ,
primary key ( stuid )
)
insert into student (stuid, lname, fname, age, sex, major, advisor,
city_code) values ( 1001, Smith, Linda, 18, F, 600, 1121, BAL);
```

Question: How many students are there for each major?
Answer: "How many students" returns 1 column. "How many students are there for each major" returns 2 columns. The question returns the number of students for each major. So the question returns 2 columns.

```
CREATE TABLE accounts (
account_id number ,
customer_id number ,
date_account_opened time ,
account_name text ,
other_account_details text ,
primary key ( account_id ) ,
foreign key ( customer_id ) references customers ( customer_id )
)
insert into accounts (account_id, customer_id, date_account_opened,
account_name, other_account_details) values (1, 6, 2016-07-30 22:22:24, 338,
 Regular);

CREATE TABLE customers (
customer_id number ,
customer_first_name text ,
customer_last_name text ,
customer_address text ,
customer_phone text ,
customer_email text ,
other_customer_details text ,
primary key ( customer_id )
)
insert into customers (customer_id, customer_first_name, customer_last_name,
 customer_address, customer_phone, customer_email, other_customer_details)
values (1, Aniyah, Feest, 55975 Theodore Estates Lake Brody, VT 57078, (673)
872-5338, fahey.dorian@example.com, NULL);
```

Question: What are the full names of customers who have accounts?
Answer: "full names" include the first and the last name. "What are the full names" returns 2 columns. The question returns the full names of customers who have accounts. So the question returns 2 columns.

Question: What are the ids, date opened, name, and other details for all accounts?
Answer: "What are the ids" returns 1 column. "What are the date opened" returns 1 column. "What are the ids, date opened" returns 2 columns. "What are the name" returns 1 column. "What are the ids, date opened, name" returns 3 columns. "What are the other details" returns 1 column. "What are the ids, date opened, name, and other details" returns 4 columns. The question returns the ids, date opened, name, and other details for all accounts. So the question returns 4 columns.

```
CREATE TABLE station (
id number ,
name text ,
lat number ,
long number ,
```

```
dock_count number ,
city text ,
installation_date text ,
primary key ( id )
)
insert into station (id, name, lat, long, dock_count, city,
installation_date) values (2, San Jose Diridon Caltrain Station,
37.329732,-1.21901782000000011405e+02,27, San Jose, 8/6/2013);

CREATE TABLE status (
station_id number ,
bikes_available number ,
docks_available number ,
time text ,
primary key ( ) ,
foreign key ( station_id ) references station ( id )
)
insert into status (station_id, bikes_available, docks_available,time)
values (3,12,3, 2015-06-02 12:46:02);

CREATE TABLE trip (
id number ,
duration number ,
bike_id number ,
start_date time ,
start_station_name text ,
start_station_id number ,
primary key ( id )
)
insert into trip (id, duration, bike_id, start_date, start_station_name,
start_station_id) values (900504, 384, 454, 8/21/2015 17:03, Howard at 2nd,
63);
```

Question: What is the name of every city that has at least 15 stations and how many stations does it have?
Answer: "What is the name" returns 1 column. "What is the name of every city that has at least 15 stations" returns 1 column. "What is the name of every city that has at least 15 stations and how many stations does it have" returns 2 columns. The question returns the name and the number of stations of every city that has at least 15 stations. So the question returns 2 columns.

Question: Which start station had the most trips starting from August? Give me the name and id of the station.
Answer: "Give me the name" returns 1 column. "Give me the id" returns 1 column. "Give me the name and id" returns 2 columns. The question returns the name and id of the station that had the most trips starting from August. So the question returns 2 columns.

```
CREATE TABLE user_profiles (
uid number ,
name text ,
followers number ,
primary key ( uid )
)
```

Question: List the name and number of followers for each user, and sort the results by the number of followers in descending order.
Answer: "List the name" returns 1 column. "List the number of followers" returns 1 column. "List the name and number of followers" returns 2 columns. "List the name and number of followers for each user" returns 2 columns. "List the name and number of followers for each user, and sort the results by

```
  the number of followers in descending order" returns 2 columns. The
question returns the name and number of followers for each user in the
descending order of the number of followers. So the question returns 2
columns.

CREATE TABLE race (
race_id number ,
name text ,
date text ,
track_id number ,
primary key ( race_id ) ,
foreign key (track_id ) references track ( track_id )
)
insert into race (race_id, name, date, track_id) values (1, Rolex 24 At
Daytona, January 26 January 27, 1);

CREATE TABLE track (
track_id number ,
name text ,
primary key ( track_id )
)
insert into track (track_id, name) values (1, Auto Club Speedway);

Question: Show the name and date for each race and its track name.
Answer: "Show the name" returns 1 column. "Show the date" returns 1 column.
"Show the name and date" returns 2 columns. "Show the name and date for each
 race" returns 2 columns. "Show the name and date for each race and its
track name" returns 3 columns. The question returns the name, date and track
 name for each race. So the question returns 3 columns.

CREATE TABLE station (
id number ,
network_name text ,
services text ,
primary key ( id )
)
insert into station (id, network_name, services) values (1, Amersham,
Metropolitan line and Chiltern Railways);

CREATE TABLE routes (
train_id number ,
station_id number ,
primary key ( train_id ) ,
foreign key ( station_id ) references station ( id )
)
insert into route (train_id, station_id) values (1,1);

Question: Find the number of trains for each station, as well as the station
 network name and services.
Answer: "Find the number of trains" returns 1 column. "Find the number of
trains for each station" returns 2 columns. "Find the number of trains for
each station, as well as the station network name" returns 2 columns. "Find
the number of trains for each station, as well as the station network name
and services" returns 3 columns. The question returns the number of trains,
the station network name and services for each station. So the question
returns 3 columns.

CREATE TABLE manufacturers (
code number ,
```

```
name text ,
primary key ( code )
)
insert into manufacturers (code, name) values (1, Sony);

CREATE TABLE products (
code number ,
name text ,
manufacturer number ,
primary key ( code ) ,
foreign key ( manufacturer ) references manufacturer ( code )
)
insert into products (code, name, manufacturer) values (1, Hard drive, 5);

Question: Find the number of products for each manufacturer, showing the
name of each company.
Answer: "Find the number of products" returns 1 column. "Find the number of
products for each manufacturer" returns 2 columns. "Find the number of
products for each manufacturer, showing the name of each company" returns 2
columns. The question returns the number of products and the company name
for each manufacturer. So the question returns 2 columns.

<insert database schemas and the new question here>
```

## H.5 PROMPT FOR SQL EXPLANATION

```
Summarize the return type of the execution.

SQL: SELECT creation FROM department GROUP BY creation ORDER BY COUNT(*)
DESC LIMIT 1

Execution:
| 1789 |
Answer: The execution of the SQL query above would return a table with 1
column. The first column, "creation" would contain the year in which a
department was created. With "GROUP BY creation ORDER BY COUNT(*) DESC", the
 table would be sorted in descending order by the number of departments
created in each year. With "LIMIT 1", only the first (i.e., highest) result
would be returned. So the SQL query returns a table with 1 column, the year
in which most departments were created.

SQL: SELECT manufacturer, AVG(price) FROM products GROUP BY manufacturer

Execution:
| 1 | 240.0 |
| 2 | 150.0 |
Answer: The execution of the SQL query above would return a table with 2
columns. The first column, "manufacturer" would contain the manufacturer
code. The second column, "AVG(price)" would contain the average price of
each manufacturer's product. So the SQL query returns a table with 2 columns
, the manufacturer code, and the average price of each manufacturer's
product.

SQL: SELECT customers.customer_name FROM customers JOIN orders ON customers.
customer_id = orders.customer_id WHERE orders.order_status = "On Road"
INTERSECT SELECT customers.customer_name FROM customers JOIN orders ON
customers.customer_id = orders.customer_id WHERE orders.order_status = "
Shipped"
```

Execution:
| George |
Answer: The execution of the SQL query above would return a table with 1 column. The first column, "customers.customer_name" would contain the customer names. With "customers JOIN orders", the table would contain the data about customers with orders. With "WHERE orders.order_status = 'On Road'", the table filters the records to only include customers who have order status "On Road". With "WHERE orders.order_status = 'Shipped'", the table filters the records to only include customers who have order status "Shipped". With "INTERSECT", the table would be filtered to only include customers who have both "On Road" and "Shipped" as order status. So the SQL query returns a table with 1 column, the customer names who have the order status "On Road" and "Shipped".

SQL: SELECT MAX(cows), MIN(cows) FROM farm

Execution:
| 3987.0 | 2407.2 |
Answer: The execution of the SQL query above would return a table with 2 columns. The first column, "MAX(cows)" would contain the maximum number of cows. The second column, "MIN(cows)" would contain the minimum number of cows. So the SQL query returns a table with 2 columns, the maximum and minimum number of cows.

SQL: SELECT AVG(rating.stars), movie.title FROM rating JOIN movie ON rating.mid = movie.mid WHERE movie.year = (SELECT MIN(year) FROM movie)

Execution:
| 4.5 | Snow White |
Answer: The execution of the SQL query above would return a table with 2 columns. The first column, "AVG(rating.stars)" would contain the average rating. The second column, "movie.title" would contain the movie title. With "rating JOIN movie", the table would contain the data about ratings and movies. With "WHERE movie.year = (SELECT MIN(year) FROM movie)", the table filters the records to only include movies with the minimum year, which means the earliest year. So the SQL query returns a table with 2 columns, the average rating and the movie title of the movie with the earliest year.

SQL: SELECT course.title, course.credits FROM classroom JOIN section ON classroom.building = section.building AND classroom.room_number = section.room_number JOIN course ON section.course_id = course.course_id WHERE classroom.capacity = (SELECT MAX(capacity) FROM classroom)

Execution:
| Accounting | 3 |
Answer: The execution of the SQL query above would return a table with 2 columns. The first column, "course.title" would contain the course title. The second column, "course.credits" would contain the course credits. With "classroom JOIN section", the table would contain the data about classrooms and sections. With "JOIN course", the table would contain the data about courses. With "WHERE classroom.capacity = (SELECT MAX(capacity) FROM classroom)", the table filters the records to only include classrooms with the maximum capacity, which means the largest capacity. So the SQL query returns a table with 2 columns, the course title and the course credits of the classroom with the largest capacity.

SQL: SELECT model_name, ram_mib FROM chip_model ORDER BY ram_mib ASC LIMIT 1;

Execution:
| X5 | 32.0 |

**Answer:** The execution of the SQL query above would return a table with 2 columns. The first column, "model_name" would contain the model name. The second column, "ram_mib" would contain the RAM size. With "ORDER BY ram_mib ASC", the table would be sorted in ascending order by RAM size. With "LIMIT 1", only the first (i.e., lowest) result would be returned. So the SQL query returns a table with 2 columns, the model name and the RAM size of the chip model with the lowest RAM size.

**SQL:** SELECT official_name, status FROM city ORDER BY population DESC LIMIT 1

**Execution:**
| Grand Falls/Grand-Sault | Town |
**Answer:** The execution of the SQL query above would return a table with 2 columns. The first column, "official_name" would contain the official name. The second column, "status" would contain the status. With "ORDER BY population DESC", the table would be sorted in descending order by population, which means from the most number of people to the least number of people. With "LIMIT 1", only the first (i.e., highest) result would be returned. So the SQL query returns a table with 2 columns, the official name and the status of the city with the largest population, which means with the most number of people.

**SQL:** SELECT AVG(budget_million), MAX(budget_million), MIN(budget_million) FROM movie WHERE year < 2000

**Execution:**
| 6.0 | 7.0 | 5.0 |
**Answer:** The execution of the SQL query above would return a table with 3 columns. The first column, "AVG(budget_million)" would contain the average budget. The second column, "MAX(budget_million)" would contain the maximum budget. The third column, "MIN(budget_million)" would contain the minimum budget. With "WHERE year < 2000", the table filters the records to only include movies before 2000. So the SQL query returns a table with 3 columns, the average, maximum, and minimum budget of movies before 2000.

**SQL:** SELECT department.department_id, department.name, COUNT(*) FROM management JOIN department ON department.department_id = management. department_id GROUP BY department.department_id HAVING COUNT(*) > 1

**Execution:**
| 2 | Treasury | 2 |
**Answer:** The execution of the SQL query above would return a table with 3 columns. The first column, "department.department_id" would contain the department ID. The second column, "department.name" would contain the department name. The third column, "COUNT(*)" would contain the number of managers. With "management JOIN department", the table would contain the data about management and departments. With "GROUP BY department. department_id", the table would be grouped by department ID. With "HAVING COUNT(*) > 1", the table filters the records to only include departments with more than 1 manager. So the SQL query returns a table with 3 columns, the department ID, the department name, and the number of managers of departments with more than 1 manager.

**SQL:** SELECT accounts.customer_id, customers.customer_first_name, customers. customer_last_name, COUNT(*) FROM accounts JOIN customers ON accounts. customer_id = customers.customer_id GROUP BY accounts.customer_id

**Execution:**
| 3 | Joesph | Schaefer | 1 |
| 6 | Vesta | Leuschke | 1 |

**Answer:** The execution of the SQL query above would return a table with 4 columns. The first column, "accounts.customer_id" would contain the customer ID. The second column, "customers.customer_first_name" would contain the customer's first name. The third column, "customers.customer_last_name" would contain the customer's last name. The fourth column, "COUNT(*)" would contain the number of accounts. With "accounts JOIN customers", the table would contain the data about accounts and customers. With "GROUP BY accounts.customer_id", the table would be grouped by customer ID. So the SQL query returns a table with 4 columns, the customer ID, the customer's first name, the customer's last name, and the number of accounts of each customer.

**SQL:** SELECT * FROM people

**Execution:**
| 1 | M | Hubert Henno | 06.10.1976 | 188.0 | 83.0 |
| 2 | M | Dominique Daquin | 10.11.1972 | 197.0 | 85.0 |
**Answer:** The execution of the SQL query above would return a table with 6 columns. "SELECT *" returns all columns. So the SQL query returns all information of people.

**SQL:** SELECT name, city, country, elevation FROM airports WHERE city = "New York"

**Execution:**
| La Guardia Airport | New York | United States | 21 |
| John F Kennedy International Airport | New York | United States | 13 |
**Answer:** The execution of the SQL query above would return a table with 4 columns. The first column, "name" would contain the airport name. The second column, "city" would contain the city name. The third column, "country" would contain the country name. The fourth column, "elevation" would contain the elevation. With "WHERE city = 'New York'", the table filters the records to only include airports in New York. So the SQL query returns a table with 4 columns, the airport name, the city name, the country name, and the elevation of airports in New York.

**SQL:** SELECT AVG(weight), MIN(weight), sex FROM people GROUP BY sex

**Execution:**
| 92.0 | 90.0 | M |
| 86.71428571428571 | 82.0 | F |
**Answer:** The execution of the SQL query above would return a table with 3 columns. The first column, "AVG(weight)" would contain the average weight. The second column, "MIN(weight)" would contain the minimum weight. The third column, "sex" would contain the gender. With "GROUP BY sex", the table would be grouped by gender. So the SQL query returns a table with 3 columns, the average weight, the minimum weight, and the gender of people for each gender.

**SQL:** SELECT COUNT(*), city FROM employees WHERE title = "IT Staff" GROUP BY city

**Execution:**
| 2 | Lethbridge |
**Answer:** The execution of the SQL query above would return a table with 2 columns. The first column, "COUNT(*)" would contain the number of employees. The second column, "city" would contain the city name. With "WHERE title = 'IT Staff'", the table filters the records to only include employees with the title "IT Staff". With "GROUP BY city", the table would be grouped by city. So the SQL query returns a table with 2 columns, the number of employees with the title "IT Staff", and the city name for each city.

```
SQL: SELECT customers.customer_name FROM customers JOIN orders ON customers.
customer_id = orders.customer_id WHERE orders.order_status = "On Road" AND
orders.order_status = "Shipped"

Execution: None
Answer: The execution of the SQL query above would return an empty table.
The first column, "customers.customer_name" would contain the customer name.
 With "customers JOIN orders", the table would contain the data about
customers with orders. In the WHERE clause, with "orders.order_status = 'On
Road'", the table filters the records to only include customers who have
order status "On Road". With "orders.order_status = 'Shipped'", the table
filters the records to only include customers who have order status "Shipped
". With "WHERE orders.order_status = 'On Road' AND orders.order_status = '
Shipped'", the table filters the records to only include customers who have
both "On Road" and "Shipped" as order status at the same time. This is not
possible, since the two are mutually exclusive. So the SQL query returns an
empty table, because the "WHERE" clause contains mutually exclusive
conditions.

SQL: SELECT origin FROM flight WHERE destination = "HONO"

Execution: None
Answer: The execution of the SQL query above would return an empty table.
The first column, "origin" would contain the origin of flights. With "WHERE
destination = 'HONO'", the table filters the records to only include flights
 whose destination is "HONO". So the SQL query returns an empty table.

<insert the new SQL here>
```

## H.6    CHAT SIMPLE FEEDBACK PROMPT (0-SHOT)

```
System: You are an expert programming assistant.
User: Write the SQL query for the given question.
<insert the database schema and the question here>
Assistant: <chatGPT Completion>
User: Does the SQL match the question? If not, generate the fixed SQL.
Assistant: <chatGPT Completion>
[...]
```

## H.7    CHAT EXPLANATION FEEDBACK PROMPT (0-SHOT)

```
System: You are an expert programming assistant.
User: Summarize the return type of the SQL execution.
<insert the database schema and the SQL here>
Assistant: <chatGPT Completion>
User: Infer the return type of the question.
<insert the question here>
Assistant: <chatGPT Completion>
User: Does the SQL match the question? If not, generate the fixed SQL.
Assistant: <chatGPT Completion>
[...]
```

# I   CODE TRANSLATION

## I.1   BASELINE PROMPT (3-SHOT)

The prompt for generating the initial Python translation is taken from (Chowdhery et al., 2022). We set temperature $\tau = 0.8$ as in (Chowdhery et al., 2022).

```
Translate the following C++ program into Python
[code]
[c++]
int add_1_to_a_given_number ( int x ) {
  int m = 1;
  while ( x & m ) {
    x = x ^ m;
    m <<= 1;
  }
  x = x ^ m;
  return x;
}
[python]
def add_1_to_a_given_number(x):
    m = 1
    while (x & m):
        x = (x ^ m)
        m <<= 1
    x = (x ^ m)
    return x
[c++]
unsigned int program_for_factorial_of_a_number ( unsigned int n ) {
  if ( n == 0 ) return 1;
  return n * program_for_factorial_of_a_number ( n - 1 );
}
[python]
def program_for_factorial_of_a_number(n):
    return (1 if ((n == 1) or (n == 0)) else (n *
program_for_factorial_of_a_number((n - 1))))
[c++]
long long int sum_pairwise_products ( int n ) {
  long long int sum = 0;
  for ( int i = 1;
  i <= n;
  i ++ ) for ( int j = i;
  j <= n;
  j ++ ) sum = sum + i * j;
  return sum;
}
[python]
def sum_pairwise_products(n):
    sm = 0
    for i in range(1, (n + 1)):
        for j in range(i, (n + 1)):
            sm = (sm + (i * j))
    return sm
[c++]
<insert C++ program here>
[python]
```

## I.2 BASELINE + EXPL. PROMPT (3-SHOT)

```
Explain the code line by line, and then translate the following C++ program
into Python
[c++]
int add_1_to_a_given_number ( int x ) {
  int m = 1;
  while ( x & m ) {
    x = x ^ m;
    m <<= 1;
  }
  x = x ^ m;
  return x;
}
[/c++]
[explanation]
The code is an implementation of adding 1 to a given number.
int m = 1
Using the variable m as a mask, the initial value of m is 1.

while ( x & m ) {
    x = x ^ m;
    m <<= 1;
}
In the while loop, if the current digit of the given number is 1, then it
will be flipped to 0, and the mask will be shifted to the next digit. This
iteration will be executed until the first 0 of the number is met.

x = x ^ m;
The first 0 will be flipped to 1. Thus, 1 will be added to the given number.

return x;
Finally, the function returns the incremented number.
[/explanation]
[python]
def add_1_to_a_given_number(x):
    m = 1
    while (x & m):
        x = (x ^ m)
        m <<= 1
    x = (x ^ m)
    return x
[/python]
[c++]
unsigned int program_for_factorial_of_a_number ( unsigned int n ) {
  if ( n == 0 ) return 1;
  return n * program_for_factorial_of_a_number ( n - 1 );
}
[/c++]
[explanation]
The code is an implementation of calculating the factorial of a number.

if ( n == 0 ) return 1;
The function is defined recursively. When the given number is equal to 0,
the result of the factorial is 1.

return n * program_for_factorial_of_a_number ( n - 1 );
Otherwise, the result of the factorial is the product of the given number
and the (given number minus 1) factorial.
[/explanation]
[python]
def program_for_factorial_of_a_number(n):
```

```
    return (1 if ((n == 1) or (n == 0)) else (n *
program_for_factorial_of_a_number((n - 1))))
[/python]
[c++]
long long int sum_pairwise_products ( int n ) {
  long long int sum = 0;
  for ( int i = 1;
  i <= n;
  i ++ ) for ( int j = i;
  j <= n;
  j ++ ) sum = sum + i * j;
  return sum;
}
[/c++]
[explanation]
The code is an implementation of summing the products of all the possible
pairs.

long long int sum = 0;
The variable sum is used to record the result.

for ( int i = 1;
  i <= n;
  i ++ ) for ( int j = i;
  j <= n;
  j ++ )
There are two nested loops. The outer loop iterates until i (the index of
the first number of a pair) is equal to the given number. The inner loop
iterates until j (the index of the second number of the pair) is equal to
the given number.

sum = sum + i * j;
When the two loops are executed, the product of the pair will be calculated
and then added to the variable sum.

return sum;
Finally, the function returns the sum.
[/explanation]
[python]
def sum_pairwise_products(n):
    sm = 0
    for i in range(1, (n + 1)):
        for j in range(i, (n + 1)):
            sm = (sm + (i * j))
    return sm
[/python]
[c++]
<insert C++ program here>
[/c++]
[explanation]
```

## I.3 SIMPLE FEEDBACK PROMPT (2-SHOT)

```
Below are C++ programs with incorrect Python translations. Correct the
translations using the provided feedback.
[c++]
unsigned int program_for_factorial_of_a_number ( unsigned int n ) {
  if ( n == 0 ) return 1;
  return n * program_for_factorial_of_a_number ( n - 1 );
```

```
}
[/c++]
[python]
def program_for_factorial_of_a_number(n):
    return (1 if ((n == 1)) else (n * program_for_factorial_of_a_number((n -
 1))))
[/python]
The above Python translation does not do the same thing as the C++ code.
Correct the Python translation.
[python]
def program_for_factorial_of_a_number(n):
    return (1 if ((n == 0)) else (n * program_for_factorial_of_a_number((n -
 1))))
[/python]
[c++]
long long int sum_pairwise_products ( int n ) {
  long long int sum = 0;
  for ( int i = 1;
  i <= n;
  i ++ ) for ( int j = i;
  j <= n;
  j ++ ) sum = sum + i * j;
  return sum;
}
[/c++]
[python]
def sum_pairwise_products(n):
    sum = 0
    for i in range(n):
        for j in range(i,((n + 1))):
            sum = (sum + (i * j))
    return sum
[/python]
The above Python translation does not do the same thing as the C++ code.
Correct the Python translation.
[python]
def sum_pairwise_products(n):
    sum = 0
    for i in range(1,((n + 1))):
        for j in range(i,((n + 1))):
            sum = (sum + (i * j))
    return sum
[/python]
[c++]
<insert C++ program here>
[/c++]
[python]
<insert original Python translation here>
[/python]
The above Python translation does not do the same thing as the C++ code.
Correct the Python translation.
[python]
```

## I.4  UNIT TEST FEEDBACK PROMPT (2-SHOT)

```
Below are C++ programs with incorrect Python translations. Correct the
translations using the provided feedback.
[c++]
unsigned int program_for_factorial_of_a_number ( unsigned int n ) {
```

```
    if ( n == 0 ) return 1;
    return n * program_for_factorial_of_a_number ( n - 1 );
}
[/c++]
[python]
def program_for_factorial_of_a_number(n):
    return (1 if ((n == 1)) else (n * program_for_factorial_of_a_number((n -
1))))
[/python]
The Python translation does not do the same thing as the C++ code. These are
 the results of one failed unit test that tests whether the Python
translation's outputs match the C++ program's outputs:
Failed: assert program_for_factorial_of_a_number(0) == 1
Actual Result: RecursionError: maximum recursion depth exceeded in
comparison
Correct the Python translation.
[python]
def program_for_factorial_of_a_number(n):
    return (1 if ((n == 0)) else (n * program_for_factorial_of_a_number((n -
1))))
[/python]
[c++]
long long int sum_pairwise_products ( int n ) {
  long long int sum = 0;
  for ( int i = 1;
  i <= n;
  i ++ ) for ( int j = i;
  j <= n;
  j ++ ) sum = sum + i * j;
  return sum;
}
[/c++]
[python]
def sum_pairwise_products(n):
    sum = 0
    for i in range(n):
        for j in range(i,((n + 1))):
            sum = (sum + (i * j))
    return sum
[/python]
The Python translation does not do the same thing as the C++ code. These are
 the results of one failed unit test that tests whether the Python
translation's outputs match the C++ program's outputs:
Failed: assert sum_pairwise_products(3) == 25
Actual Result: 16
Correct the Python translation.
[python]
def sum_pairwise_products(n):
    sum = 0
    for i in range(1,((n + 1))):
        for j in range(i,((n + 1))):
            sum = (sum + (i * j))
    return sum
[/python]
[c++]
<insert C++ program here>
[/c++]
[python]
<insert original Python translation here>
[/python]
```

## I.5   UNIT TEST + EXPL. FEEDBACK PROMPT (2-SHOT)

```
Below are C++ programs with incorrect Python translations. Explain the
original code, then explain the translations line by line and correct them
using the provided feedback.
[c++]
unsigned int program_for_factorial_of_a_number ( unsigned int n ) {
  if ( n == 0 ) return 1;
  return n * program_for_factorial_of_a_number ( n - 1 );
}
[/c++]
[explanation]
The code is an implementation of calculating the factorial of a number.

if ( n == 0 ) return 1;
The function is defined recursively. When the given number is equal to 0,
the result of the factorial is 1.

return n * program_for_factorial_of_a_number ( n - 1 );
Otherwise, the result of the factorial is the product of the given number
and the (given number minus 1) factorial.
[/explanation]
[python]
def program_for_factorial_of_a_number(n):
    return (1 if ((n == 1)) else (n * program_for_factorial_of_a_number((n -
 1))))
[/python]
Explain the Python translation line by line.
[explanation]
The code is an implementation of calculating the factorial of a number.

return (1 if ((n == 1)) else (n * program_for_factorial_of_a_number((n - 1))
))

The function is defined recursively. When the given number is equal to 1,
the result of the factorial is 1. Otherwise, the result of the factorial is
the product of the given number and the (given number minus 1) factorial.
[/explanation]
The Python translation does not do the same thing as the C++ code. These are
 the results of one failed unit test that tests whether the Python
translation's outputs match the C++ program's outputs:
Failed: assert program_for_factorial_of_a_number(0) == 1
Actual Result: RecursionError: maximum recursion depth exceeded in
comparison
Correct the translation.
[python]
def program_for_factorial_of_a_number(n):
    return (1 if ((n == 1) or (n == 0)) else (n *
program_for_factorial_of_a_number((n - 1))))
[/python]
[c++]
long long int sum_pairwise_products ( int n ) {
  long long int sum = 0;
  for ( int i = 1;
  i <= n;
  i ++ ) for ( int j = i;
  j <= n;
  j ++ ) sum = sum + i * j;
  return sum;
}
[/c++]
```

[explanation]
The code is an implementation of summing the products of all the possible pairs.

```
long long int sum = 0;
```
The variable sum is used to record the result.

```
for ( int i = 1;
  i <= n;
  i ++ ) for ( int j = i;
  j <= n;
  j ++ )
```
There are two nested loops. The outer loop iterates until i (the index of the first number of a pair) is equal to the given number. The inner loop iterates until j (the index of the second number of the pair) is equal to the given number.

```
sum = sum + i * j;
```
When the two loops are executed, the product of the pair will be calculated and then added to the variable sum.

```
return sum;
```
Finally, the function returns the sum.
[/explanation]
[python]
```python
def sum_pairwise_products(n):
    sum = 0
    for i in range(n):
        for j in range(i,((n + 1))):
            sum = (sum + (i * j))
    return sum
```
[/python]
Explain the Python translation line by line.
[explanation]
The code is an implementation of summing the products of all the possible pairs.

```
sum = 0;
```
The variable sum is used to record the result.

```
for i in range(n):
    for j in range(i,((n + 1))):
```
There are two nested loops. The outer loop iterates from i = 0 until i is equal to n-1. The inner loop iterates from j = i until j is equal to n.

```
sum = (sum + (i * j))
```
When the two loops are executed, the product of the pair will be calculated and then added to the variable sum.

```
return sum
```
Finally, the function returns the sum.
[/explanation]
The Python translation does not do the same thing as the C++ code. These are the results of one failed unit test that tests whether the Python translation's outputs match the C++ program's outputs:
Failed: assert sum_pairwise_products(3) == 25
Actual Result: 16
Correct the translation.
[python]
```python
def sum_pairwise_products(n):
    sm = 0
    for i in range(1, (n + 1)):
        for j in range(i, (n + 1)):
```

```
            sm = (sm + (i * j))
    return sm
[/python]
[c++]
<insert C++ program here>
[/c++]
[explanation]
<insert explanation of C++ program here>
[/explanation]
[python]
<insert original Python translation here>
[/python]
```

I.6   UNIT TEST + TRACE FEEDBACK PROMPT (2-SHOT)

```
Below are C++ programs with incorrect Python translations. Using the
provided feedback, trace through the execution of the translations to
determine what needs to be fixed, and correct the translations.
[c++]
unsigned int program_for_factorial_of_a_number ( unsigned int n ) {
  if ( n == 0 ) return 1;
  return n * program_for_factorial_of_a_number ( n - 1 );
}
[/c++]
[python]
def program_for_factorial_of_a_number(n):
    return (1 if ((n == 1)) else (n * program_for_factorial_of_a_number((n -
 1))))
[/python]
The Python translation does not do the same thing as the C++ code. These are
 the results of one failed unit test that tests whether the Python
translation's outputs match the C++ program's outputs:
Failed: assert program_for_factorial_of_a_number(0) == 1
Actual Result: RecursionError: maximum recursion depth exceeded in
comparison
Trace the execution of the Python translation for
program_for_factorial_of_a_number(0).
[trace]
Here's how the execution proceeds:

1. The function is called with 'n' equals 0.
2. The function checks if 'n' equals 1. Since 'n' is 0, this condition is '
False'.
3. The function then calls itself with 'n' equals -1 ('n' - 1).
4. Steps 2 and 3 repeat infinitely because there is no base case to exit the
 recursion when 'n' equals 0.

To fix this issue, the base case should be changed to 'n' equals 0.
[/trace]
Correct the Python translation.
[python]
def program_for_factorial_of_a_number(n):
    return (1 if ((n == 0)) else (n * program_for_factorial_of_a_number((n -
 1))))
[/python]
[c++]
long long int sum_pairwise_products ( int n ) {
  long long int sum = 0;
  for ( int i = 1;
```

```
   i <= n;
   i ++ ) for ( int j = i;
   j <= n;
   j ++ ) sum = sum + i * j;
   return sum;
}
[/c++]
[python]
def sum_pairwise_products(n):
    sum = 0
    for i in range(n):
        for j in range(i,((n + 1))):
            sum = (sum + (i * j))
    return sum
[/python]
The Python translation does not do the same thing as the C++ code. These are
 the results of one failed unit test that tests whether the Python
translation's outputs match the C++ program's outputs:
Failed: assert sum_pairwise_products(3) == 25
Actual Result: 16
Trace the execution of the Python translation for sum_pairwise_products(3).
[trace]
Here's how the execution proceeds:

1. The function is called with 'n' equals 3.
2. The 'i' loop runs from 0 to 2. In C++ code, the loop starts from 1. This
is because in Python, the 'range()' function starts counting from 0, while
in C++, the 'for' loop starts counting from the initial value which is 1 in
your case.
3. The 'j' loop runs from 'i' to 3, which is correct.
4. The product of 'i' and 'j' is added to the sum. Because 'i' starts from 0
 in Python, the sum is less than expected.

To fix the issue, the 'i' loop should start from 1, not 0. The Python 'range
()' function should be 'range(1, n + 1)'. This way, 'i' will start from 1,
just like in the original C++ code.
[/trace]
Correct the Python translation.
[python]
def sum_pairwise_products(n):
    sm = 0
    for i in range(1, (n + 1)):
        for j in range(i, (n + 1)):
            sm = (sm + (i * j))
    return sm
[/python]
[c++]
<insert C++ program here>
[/c++]
[python]
<insert original Python translation here>
[/python]
```

## I.7    CHAT SIMPLE FEEDBACK PROMPT (0-SHOT)

```
System: You are an expert programming assistant.
User: Translate the C++ code into Python code. Please respond with code only
 (with the code inside a Markdown code block).
<insert C++ program here>
```

```
Assistant: <chatGPT Completion>
User: The Python translation does not do the same thing as the C++ code.
Please correct the Python translation.
Assistant: <chatGPT Completion>
User: The Python translation does not do the same thing as the C++ code.
Please correct the Python translation.
[...]
```

## I.8 CHAT UNIT TEST FEEDBACK PROMPT (0-SHOT)

The initial part of the prompt is the same as that of the TransCoder Chat Simple Feedback prompt in Appendix I.7. The only difference is the feedback message.

```
System: You are an expert programming assistant.
User: Translate the C++ code into Python code. Please respond with code only
  (with the code inside a Markdown code block).
<insert C++ program here>
Assistant: <chatGPT Completion>
User: The Python translation does not do the same thing as the C++ code.
These are the results of one failed unit test that tests whether the Python
translation's outputs match the C++ program's outputs:
<insert one failed unit test and its execution result here>
Correct the Python translation.
Assistant: <chatGPT Completion>
User: The Python translation does not do the same thing as the C++ code.
These are the results of one failed unit test that tests whether the Python
translation's outputs match the C++ program's outputs:
<insert one failed unit test and its execution result here>
Correct the Python translation.
[...]
```

## I.9 CHAT UNIT TEST + EXPL. FEEDBACK PROMPT (0-SHOT)

```
System: You are an expert programming assistant.
User: Explain the C++ code line by line:
<insert C++ program here>
Assistant: <chatGPT Completion>
User: Translate the above C++ code into Python code. Please respond with
code only (with the code inside a Markdown code block).
Assistant: <chatGPT Completion>
User: Explain the Python translation line by line.
Assistant: <chatGPT Completion>
User: The Python translation does not do the same thing as the C++ code.
These are the results of one failed unit test that tests whether the Python
translation's outputs match the C++ program's outputs:
<insert one failed unit test and its execution result here>
Correct the Python translation.
Assistant: <chatGPT Completion>
User: Explain the Python translation line by line.
[...]
```

### I.10    CHAT UNIT TEST + TRACE FEEDBACK PROMPT (0-SHOT)

```
System: You are an expert programming assistant.
User: Translate the C++ code into Python code. Please respond with code only
  (with the code inside a Markdown code block).
<insert C++ program here>
Assistant: <chatGPT Completion>
User:
The Python translation does not do the same thing as the C++ code. These are
 the results of one failed unit test that tests whether the Python
translation's outputs match the C++ program's outputs:
<insert one failed unit test and its execution result here>
Trace the execution of <function call of failed unit test.
Assistant: <chatGPT Completion>
User: Correct the Python translation.
Assistant: <chatGPT Completion>
User:
The Python translation does not do the same thing as the C++ code. These are
 the results of one failed unit test that tests whether the Python
translation's outputs match the C++ program's outputs:
<insert one failed unit test and its execution result here>
Trace the execution of <function call of failed unit test.
Assistant: <chatGPT Completion>
User: Correct the Python translation.
Assistant: <chatGPT Completion>
[...]
```

## J    TEXT-TO-PYTHON GENERATION

### J.1    BASELINE PROMPT (3-SHOT)

The prompt for generating initial Python code is taken from (Ni et al., 2023).

```
# Write Python function to complete the task and pass the assertion tests.

### Task Start ###
# These are the assertions for your function:
assert similar_elements((3, 4, 5, 6),(5, 7, 4, 10)) == (4, 5)

""" Write a function to find the similar elements from the given two tuple
lists. """
def similar_elements(test_tup1, test_tup2):
    res = tuple(set(test_tup1) & set(test_tup2))
    return (res)
### Task End ###

### Task Start ###
# These are the assertions for your function:
assert is_not_prime(2) == False

""" Write a python function to identify non-prime numbers. """
import math
def is_not_prime(n):
    result = False
    for i in range(2,int(math.sqrt(n)) + 1):
        if n % i == 0:
            result = True
    return result
```

```
### Task End ###

### Task Start ###
# These are the assertions for your function:
assert heap_queue_largest( [25, 35, 22, 85, 14, 65, 75, 22, 58],3)==[85, 75,
 65]

""" Write a function to find the largest integers from a given list of
numbers using heap queue algorithm. """
import heapq as hq
def heap_queue_largest(nums,n):
    largest_nums = hq.nlargest(n, nums)
    return largest_nums
### Task End ###

### Task Start ###
# These are the assertions for your function:
<insert assertions and problem description here>
```

## J.2   SIMPLE FEEDBACK PROMPT (6-SHOT)

```
# Write Python function to complete the task and pass the assertion tests.

### Task Start ###
# These are the assertions for your function:
assert count_ways(2) == 3

""" Write a function to find the number of ways to fill it with 2 x 1
dominoes for the given 3 x n board. """
def count_ways(n):
    if n == 0:
        return 1
    if n == 1:
        return 1
    if n == 2:
        return 3
    return count_ways(n-1) + count_ways(n-2)

Feedback: The code above is wrong. Please fix it.

def count_ways(n):
  A = [0] * (n + 1)
  B = [0] * (n + 1)
  A[0] = 1
  A[1] = 0
  B[0] = 0
  B[1] = 1
  for i in range(2, n+1):
    A[i] = A[i - 2] + 2 * B[i - 1]
    B[i] = A[i - 1] + B[i - 2]
  return A[n]

Feedback: The code above is correct.

### Task End ###

### Task Start ###
# These are the assertions for your function:
```

```
assert differ_At_One_Bit_Pos(15,8) == False

""" Write a python function to check whether the two numbers differ at one
bit position only or not. """
def differ_At_One_Bit_Pos(lhs,rhs):
    if (lhs - rhs) == 0 or (lhs - rhs) == 1:
        return True
    return False

Feedback: The code above is wrong. Please fix it.

def differ_At_One_Bit_Pos(a,b):
  x = a ^ b
  return x and (not(x & (x - 1)))

Feedback: The code above is correct.

### Task End ###

### Task Start ###
# These are the assertions for your function:
assert find_char_long('Please move back to stream') == ['Please', 'move', '
back', 'stream']

""" Write a function to find all words which are at least 4 characters long
in a string by using regex. """
def find_char_long(text):
  return (re.findall(r"\b\w{4,}\b", text))

Feedback: The code above is wrong. Please fix it.

import re
def find_char_long(text):
  return (re.findall(r"\b\w{4,}\b", text))

Feedback: The code above is correct.

### Task End ###

### Task Start ###
# These are the assertions for your function:
assert square_nums([1, 2, 3, 4, 5, 6, 7, 8, 9, 10])==[1, 4, 9, 16, 25, 36,
49, 64, 81, 100]

""" Write a function to find squares of individual elements in a list using
lambda function. """
def square_nums(nums):
  square_nums = list(map(lambda x: x ** 2, nums))
  return square_nums

Feedback: The code above is correct.

### Task End ###

### Task Start ###
# These are the assertions for your function:
assert find_Rotations("aaaa") == 1

""" Write a python function to find the minimum number of rotations required
 to get the same string. """
def find_Rotations(s):
    if len(s) == 1:
```

```
        return 1
    elif len(s) == 0:
        return 0
    else:
        l = len(s)
        min = len(s)
        for i, c in enumerate(s):
            if c == s[0]:
                temp = i
                if temp < min:
                    min = temp
        return min

Feedback: The code above is wrong. Please fix it.

def find_Rotations(s):
  n = len(s)
  for i in range(1, n + 1):
    if s[i:n] + s[0:i] == s:
      return i
  return n

Feedback: The code above is correct.

### Task End ###

### Task Start ###
# These are the assertions for your function:
assert small_nnum([10, 20, 50, 70, 90, 20, 50, 40, 60, 80, 100],2)==[10,20]

"""  Write a function to get the n smallest items from a dataset. """
def small_nnum(lst,n):
  lst = sorted(lst)
  lst = lst[:n]
  return lst

Feedback: The code above is correct.

### Task End ###

### Task Start ###
# These are the assertions for your function:
<insert assertions and problem description here>
<insert original code here>
```

### J.3   UNIT TEST FEEDBACK PROMPT (6-SHOT)

```
# Write Python function to complete the task and pass the assertion tests.

### Task Start ###
# These are the assertions for your function:
assert count_ways(2) == 3

""" Write a function to find the number of ways to fill it with 2 x 1
dominoes for the given 3 x n board. """
def count_ways(n):
    if n == 0:
        return 1
```

```
        if n == 1:
            return 1
        if n == 2:
            return 3
        return count_ways(n-1) + count_ways(n-2)
```

Feedback: With the above function, count_ways(2) == 3. The assertion is "
assert count_ways(2) == 3". So the code passes the assertion. The code above
 is wrong. Please fix it.

```
def count_ways(n):
  A = [0] * (n + 1)
  B = [0] * (n + 1)
  A[0] = 1
  A[1] = 0
  B[0] = 0
  B[1] = 1
  for i in range(2, n+1):
    A[i] = A[i - 2] + 2 * B[i - 1]
    B[i] = A[i - 1] + B[i - 2]
  return A[n]
```

Feedback: With the above function, count_ways(2) = 3. The assertion is "
assert count_ways(2) == 3". So the code passes the assertion. The code above
 is correct.

```
### Task End ###

### Task Start ###
# These are the assertions for your function:
assert differ_At_One_Bit_Pos(15,8) == False

""" Write a python function to check whether the two numbers differ at one
bit position only or not. """
def differ_At_One_Bit_Pos(lhs,rhs):
    if (lhs - rhs) == 0 or (lhs - rhs) == 1:
        return True
    return False
```

Feedback: With the above function, differ_At_One_Bit_Pos(15,8) == False. The
 assertion is "assert differ_At_One_Bit_Pos(15,8) == False". So the code
passes the assertion. The code above is wrong. Please fix it.

```
def differ_At_One_Bit_Pos(a,b):
  x = a ^ b
  return x and (not(x & (x - 1)))
```

Feedback: With the above function, differ_At_One_Bit_Pos(15,8) == False. The
 assertion is "assert differ_At_One_Bit_Pos(15,8) == False". So the code
passes the assertion. The code above is correct.

```
### Task End ###

### Task Start ###
# These are the assertions for your function:
assert find_char_long('Please move back to stream') == ['Please', 'move', '
back', 'stream']

""" Write a function to find all words which are at least 4 characters long
in a string by using regex. """
def find_char_long(text):
  return (re.findall(r"\b\w{4,}\b", text))
```

Feedback: With the above function, find_char_long('Please move back to stream') returns the following error:
```
"""
NameError: name 're' is not defined
"""
```
So the code does not pass the assertion. Please fix it.

```python
import re
def find_char_long(text):
  return (re.findall(r"\b\w{4,}\b", text))
```

Feedback: With the above function, find_char_long('Please move back to stream') == ['Please', 'move', 'back', 'stream']. The assertion is "assert find_char_long('Please move back to stream') == ['Please', 'move', 'back', 'stream']". So the code passes the assertion. The code above is correct.

```
### Task End ###

### Task Start ###
# These are the assertions for your function:
assert square_nums([1, 2, 3, 4, 5, 6, 7, 8, 9, 10])==[1, 4, 9, 16, 25, 36, 49, 64, 81, 100]

""" Write a function to find squares of individual elements in a list using lambda function. """
def square_nums(nums):
  square_nums = list(map(lambda x: x ** 2, nums))
  return square_nums
```

Feedback: With the above function, square_nums([1, 2, 3, 4, 5, 6, 7, 8, 9, 10])==[1, 4, 9, 16, 25, 36, 49, 64, 81, 100]. The assertion is "assert square_nums([1, 2, 3, 4, 5, 6, 7, 8, 9, 10])==[1, 4, 9, 16, 25, 36, 49, 64, 81, 100]". So the code passes the assertion. The code above is correct.

```
### Task End ###

### Task Start ###
# These are the assertions for your function:
assert find_Rotations("aaaa") == 1

""" Write a python function to find the minimum number of rotations required
 to get the same string. """
def find_Rotations(s):
    if len(s) == 1:
        return 1
    elif len(s) == 0:
        return 0
    else:
        l = len(s)
        min = len(s)
        for i, c in enumerate(s):
            if c == s[0]:
                temp = i
                if temp < min:
                    min = temp
        return min
```

Feedback: With the above function, find_Rotations("aaaa") == 0. The assertion is "find_Rotations("aaaa") == 1". So the code does not pass the assertion. Please fix it.

```
def find_Rotations(s):
  n = len(s)
  for i in range(1, n + 1):
    if s[i:n] + s[0:i] == s:
      return i
  return n
```

Feedback: With the above function, find_Rotations("aaaa") == 1. The
assertion is "find_Rotations("aaaa") == 1". So the code passes the assertion
. The code above is correct.

```
### Task End ###

### Task Start ###
# These are the assertions for your function:
assert small_nnum([10, 20, 50, 70, 90, 20, 50, 40, 60, 80, 100],2)==[10,20]

"""  Write a function to get the n smallest items from a dataset. """
def small_nnum(lst,n):
  lst = sorted(lst)
  lst = lst[:n]
  return lst
```

Feedback: With the above function, small_nnum([10, 20, 50, 70, 90, 20, 50,
40, 60, 80, 100],2)==[10,20]. The assertion is "small_nnum([10, 20, 50, 70,
90, 20, 50, 40, 60, 80, 100],2)==[10,20]". So the code passes the assertion.
 The code above is correct.

```
### Task End ###

### Task Start ###
# These are the assertions for your function:
<insert assertions and problem description here>
<insert original code here>
```

## J.4   UNIT TEST + EXPL. FEEDBACK PROMPT (3-SHOT)

```
# Write Python function to complete the task and pass the assertion tests.

### Task Start ###
# These are the assertions for your function:
assert find_char_long('Please move back to stream') == ['Please', 'move', '
back', 'stream']

""" Write a function to find all words which are at least 4 characters long
in a string by using regex. """
def find_char_long(text):
  return (re.findall(r"\b\w{4,}\b", text))
```

Here is a line-by-line explanation of the code:
'def find_char_long(text):': This line defines a function named '
find_char_long' that takes a single argument, 'text'. 'text' represents the
string whose words are to be extracted.

'return (re.findall(r"\b\w{4,}\b", text))': This line uses the 're.findall()
' function to extract all words from the input string that are at least 4
characters long. The regular expression 'r"\b\w{4,}\b"' matches all words
that are at least 4 characters long. The '\b' matches the boundary between a

 word character and a non-word character. The '\w' matches any word
character (a letter, digit, or underscore). The '{4,}' matches the preceding
 element at least 4 times. The '\b' matches the boundary between a word
character and a non-word character.

Feedback: With the above function, find_char_long('Please move back to
stream') returns the following error:
"""
NameError: name 're' is not defined
"""
So the code does not pass the assertion. Please fix it.

```python
import re
def find_char_long(text):
  return (re.findall(r"\b\w{4,}\b", text))
```

Here is a line-by-line explanation of the code:
'import re': This line imports the 're' module.

'def find_char_long(text):': This line defines a function named '
find_char_long' that takes a single argument, 'text'. 'text' represents the
string whose words are to be extracted.

'return (re.findall(r"\b\w{4,}\b", text))': This line uses the 're.findall()
' function to extract all words from the input string that are at least 4
characters long. The regular expression 'r"\b\w{4,}\b"' matches all words
that are at least 4 characters long. The '\b' matches the boundary between a
 word character and a non-word character. The '\w' matches any word
character (a letter, digit, or underscore). The '{4,}' matches the preceding
 element at least 4 times. The '\b' matches the boundary between a word
character and a non-word character.

Feedback: With the above function, find_char_long('Please move back to
stream') == ['Please', 'move', 'back', 'stream']. The assertion is "assert
find_char_long('Please move back to stream') == ['Please', 'move', 'back', '
stream']". So the code passes the assertion. The code above is correct.

```
### Task End ###

### Task Start ###
# These are the assertions for your function:
assert count_ways(2) == 3

""" Write a function to find the number of ways to fill it with 2 x 1
dominoes for the given 3 x n board. """
def count_ways(n):
    if n == 0:
        return 1
    if n == 1:
        return 1
    if n == 2:
        return 3
    return count_ways(n-1) + count_ways(n-2)
```

Here is a line-by-line explanation of the code:
'def count_ways(n):': This line defines a function named 'count_ways' that
takes a single argument, 'n'. 'n' represents the width of the 3 x n board.

'if n == 0:': This line checks if n is equal to 0. In other words, it checks
 if the width of the board is 0. If the width of the board is 0, there is
only one way to fill it: by not placing any dominoes.

'return 1': If the condition in line 2 is true (i.e., n is 0), the function returns 1, indicating that there is only one way to fill the 3 x 0 board.

'if n == 1:': This line checks if n is equal to 1. In other words, it checks if the width of the board is 1. A 3 x 1 board cannot be filled with 2 x 1 dominoes, so there's only one way to fill it: by not placing any dominoes.

'return 1': If the condition in line 4 is true (i.e., n is 1), the function returns 1, indicating that there is only one way to fill the 3 x 1 board.

'if n == 2:': This line checks if n is equal to 2. In other words, it checks if the width of the board is 2. A 3 x 2 board can be filled with 2 x 1 dominoes in three ways: (1) three horizontal dominoes, (2) one vertical domino and two horizontal dominoes, and (3) two vertical dominoes and one horizontal domino.

'return 3': If the condition in line 6 is true (i.e., n is 2), the function returns 3, indicating that there are three ways to fill the 3 x 2 board.

'return count_ways(n-1) + count_ways(n-2)': If none of the conditions in lines 2, 4, and 6 are true, this line is executed. It uses a recursive approach to calculate the number of ways to fill the 3 x n board. The function calls itself twice: once with the argument n-1 and once with the argument n-2. The results of these two function calls are added together and returned as the final result.

Feedback: With the above function, count_ways(2) == 3. The assertion is " assert count_ways(2) == 3". So the code passes the assertion. The code above is wrong. Please fix it.

```
def count_ways(n):
  A = [0] * (n + 1)
  B = [0] * (n + 1)
  A[0] = 1
  A[1] = 0
  B[0] = 0
  B[1] = 1
  for i in range(2, n+1):
    A[i] = A[i - 2] + 2 * B[i - 1]
    B[i] = A[i - 1] + B[i - 2]
  return A[n]
```

Here is a line-by-line explanation of the code:
'def count_ways(n):': This line defines a function named 'count_ways' that takes a single argument, 'n'. 'n' represents the width of the 3 x n board.

'A = [0] * (n + 1) and B = [0] * (n + 1)': These lines create two lists, A and B, each of length n + 1 and initialized with zeros. A[i] and B[i] will store the number of ways to tile the 3 x i board while following specific rules for the last column (explained below).

'A[0] = 1, A[1] = 0, B[0] = 0, and B[1] = 1': These lines initialize the base cases for the lists A and B. A[0] and B[1] are set to 1, while A[1] and B[0] are set to 0. A[i] represents the number of ways to tile a 3 x i board with the last column fully filled, and B[i] represents the number of ways to tile a 3 x i board with the last column filled in the form of an L-shape.

'for i in range(2, n + 1):': This line starts a for loop that iterates from 2 to n, inclusive.

'A[i] = A[i - 2] + 2 * B[i - 1]': This line calculates the number of ways to tile a 3 x i board with the last column fully filled by either placing two horizontal dominoes in the last column (which corresponds to A[i - 2]) or by

```
 placing a horizontal and a vertical domino in the last column (which
corresponds to 2 * B[i - 1]).

'B[i] = A[i - 1] + B[i - 2]': This line calculates the number of ways to
tile a 3 x i board with the last column filled in the form of an L-shape by
either placing a vertical domino and a horizontal domino in the last column
(which corresponds to A[i - 1]) or by placing three horizontal dominoes in
the last column (which corresponds to B[i - 2]).

'return A[n]': This line returns the number of ways to tile the 3 x n board
with 2 x 1 dominoes, which is stored in A[n].

Feedback: With the above function, count_ways(2) = 3. The assertion is "
assert count_ways(2) == 3". So the code passes the assertion. The code above
 is correct.

### Task End ###

### Task Start ###
# These are the assertions for your function:
assert find_Rotations("aaaa") == 1

""" Write a python function to find the minimum number of rotations required
 to get the same string. """
def find_Rotations(s):
    if len(s) == 1:
        return 1
    elif len(s) == 0:
        return 0
    else:
        l = len(s)
        min = len(s)
        for i, c in enumerate(s):
            if c == s[0]:
                temp = i
                if temp < min:
                    min = temp
        return min

Here is a line-by-line explanation of the code:
'def find_Rotations(s):': This line defines a function named 'find_Rotations
' that takes a single argument, 's'. 's' represents the string whose minimum
 number of rotations is to be calculated.

'if len(s) == 1:': This line checks if the length of the input string is 1.
the minimum number of rotations required to get the same string is 1.

'return 1': If the condition in line 2 is true (i.e., length of the input
string is 1), the function returns 1.

'elif len(s) == 0:': This line checks if the length of the input string is
0. If the length of the input string is 0, the minimum number of rotations
required to get the same string is 0.

'return 0': If the condition in line 4 is true (i.e., length of the input
string is 0), the function returns 0.

'else:': If the conditions in lines 2 and 4 are not met (i.e., length of the
 input string is neither 0 nor 1), this line is executed.

'l = len(s)': This line stores the length of the input string in the
variable 'l'.
```

'min = len(s)': This line stores the length of the input string in the variable 'min'.

'for i, c in enumerate(s):': This line starts a for loop that iterates over the input string.

'if c == s[0]:': This line checks if the current character is equal to the first character of the input string.

'temp = i': This line stores the index of the current character in the variable 'temp'.

'if temp < min:': This line checks if the index of the current character is less than the value stored in the variable 'min'.

'min = temp': If the condition in line 13 is true (i.e., index of the current character is less than the value stored in the variable 'min'), this line is executed. It stores the index of the current character in the variable 'min'.

'return min': This line returns the minimum number of rotations required to get the same string.

Feedback: With the above function, find_Rotations("aaaa") == 0. The assertion is "find_Rotations("aaaa") == 1". So the code does not pass the assertion. Please fix it.

```python
def find_Rotations(s):
    n = len(s)
    for i in range(1, n + 1):
        if s[i:n] + s[0:i] == s:
            return i
    return n
```

Here is a line-by-line explanation of the code:
'def find_Rotations(s):': This line defines a function named 'find_Rotations' that takes a single argument, 's'. 's' represents the string whose minimum number of rotations is to be calculated.

'n = len(s)': This line stores the length of the input string in the variable 'n'.

'for i in range(1, n + 1):': This line starts a for loop that iterates from 1 to n, inclusive. The loop variable is i, which represents the number of rotations being tested in each iteration.

'if s[i:n] + s[0:i] == s:': This line checks if the string resulting from the concatenation of two substrings 's[i:n]' and 's[0:i]' is equal to the original string 's'. The first substring 's[i:n]' is a slice of the string from index 'i' to the end, and the second substring 's[0:i]' is a slice from the beginning of the string up to index 'i-1'. This operation essentially rotates the string by 'i' positions.

'return i': If the condition in line 5 is true (i.e., the string resulting from the concatenation of two substrings 's[i:n]' and 's[0:i]' is equal to the original string 's'), this line is executed. It returns the number of rotations required to get the same string.

'return n': This line returns the length of the string n. This means that the minimum number of rotations required to get the same string is equal to the length of the string, indicating that the string cannot be rotated to achieve the same string.

```
Feedback: With the above function, find_Rotations("aaaa") == 1. The
assertion is "find_Rotations("aaaa") == 1". So the code passes the assertion
. The code above is correct.

### Task End ###

### Task Start ###
# These are the assertions for your function:
<insert assertions and problem description here>
<insert original code here>
```

## J.5 CHAT SIMPLE FEEDBACK PROMPT (0-SHOT)

```
System: You are an expert programming assistant.
User: Complete the following task in Python:
<insert problem here (includes one unit test)>
Assistant: <chatGPT Completion>
```

Feedback for failure: If the model's prediction fails the given unit test, the next part of the prompt is:

```
User: The code above is wrong. Please fix it.
Assistant: <chatGPT Completion>
User: The code above is wrong. Please fix it.
[...]
```

Feedback for success: When the model produces a prediction that passes the given unit test (which could take place after several turns of failing the unit test), the prompt continues with a single round of feedback that allows it to make further changes to its code:

```
User: Is the code above correct? If not, please fix it.
Assistant: <chatGPT Completion>
```

The new code prediction is used only if it also passes the unit test, otherwise the original prediction which passes will be used for evaluation. (Note that after this kind of feedback, we finish SELF-DEBUGGING and do not continue with "Feedback for failure" messages even if the new prediction is incorrect.)

## J.6 CHAT UNIT TEST FEEDBACK PROMPT (0-SHOT)

The initial part of the prompt is the same as that of the MBPP Chat Simple Feedback prompt in Appendix J.5. The only difference is the feedback message.

```
System: You are an expert programming assistant.
User: Complete the following task in Python:
<insert problem here (includes one unit test)>
Assistant: <chatGPT Completion>
```

Feedback for failure: If the model's prediction fails the given unit test, the next part of the prompt is:

```
User: The code above fails the given unit test:
<insert the failed unit test and its execution result here>
Please fix the Python code.
Assistant: <chatGPT Completion>
User: The code above fails the given unit test:
<insert the failed unit test and its execution result here>
Please fix the Python code.
[...]
```

Feedback for success: Just as in the MBPP Chat Simple Feedback Prompt (0-shot) in Appendix J.5, when the model produces a prediction that passes the given unit test, the prompt continues with:

```
User: Is the code above correct? If not, please fix it.
Assistant: <chatGPT Completion>
```

The new code prediction is used only if it also passes the unit test, otherwise the original code will be used for evaluation.

### J.7 CHAT UNIT TEST + EXPL. FEEDBACK PROMPT (0-SHOT)

```
System: You are an expert programming assistant.
User: Complete the following task in Python:
<insert problem here (includes one unit test)>
Assistant: <chatGPT Completion>
```

Feedback for failure: If the model's prediction fails the given unit test, the next part of the prompt is:

```
User: Explain the Python code line by line.
Assistant: <chatGPT Completion>
User: The code above fails the given unit test:
<insert the failed unit test and its execution result here>
Please fix the Python code.
Assistant: <chatGPT Completion>
User: Explain the Python code line by line.
[...]
```

Feedback for success: Just as in the Chat Simple Feedback Prompt (0-shot) in Appendix J.5, when the model produces a prediction that passes, the prompt continues with:

```
User: Explain the Python code line by line.
Assistant: <chatGPT Completion>
User: Is the code above correct? If not, please fix it.
Assistant: <chatGPT Completion>
```

The new code prediction is used only if it also passes the unit test, otherwise the original code will be used for evaluation.

## J.8  CHAT UNIT TEST + TRACE FEEDBACK PROMPT (0-SHOT)

```
System: You are an expert programming assistant.
User: Complete the following task in Python:
<insert problem here (includes one unit test)>
Assistant: <chatGPT Completion>
User: The code above fails the given unit test:
<insert the failed unit test and its execution result here>
Trace the execution of <function call of failed unit test.
Assistant: <chatGPT Completion>
Please fix the Python code.
Assistant: <chatGPT Completion>
User: The code above fails the given unit test:
<insert the failed unit test and its execution result here>
Trace the execution of <function call of failed unit test.
Assistant: <chatGPT Completion>
Please fix the Python code.
Assistant: <chatGPT Completion>
[...]
```

