# OpenReview forum: "Teaching Large Language Models to Self-Debug"
_ICLR.cc/2024/Conference — ICLR 2024 poster_

### Official Review · Reviewer_AdYB · 2023-10-28

**Soundness:** 2 fair
**Presentation:** 2 fair
**Contribution:** 3 good
**Rating:** 6
**Confidence:** 5

**Summary:**

This paper presents a few-shot prompting based approach to help LLM's to self-correct errors made in code generation tasks. The approach works as follows:

Given a code task prompt, when LLM produces an output code, the system would first test the code on test input/output to decide if the code is correct or not. If the code is incorrect, the system call the LLM again to perform self-repair of the code, and produce a revised code as the output.

The paper various multiple ways to generate self-repair feedback: Simple (binary feedback), UT (when unit tests are available as part of task prompt), Expl (self-generated explanation of error) and Trace (execution trace when UT is available). The paper shows that different versions of feedback information have their own strength in different models and tasks; but in general, Expl shows most promising results, especially in Spider dataset.

The contribution of the paper is clear: it provides a comprehensive approach for leveraging self-repair to fix errors of code generation models, and perform ablation studies to compare their effectiveness in different tasks and different models.

**Strengths:**

* systematic study of the effectiveness of self-repair
* thorough evaluation on most recent models, tasks and variations of repair methods.

**Weaknesses:**

The paper right now has quite a big flaw in my opinion: the unfair comparison with existing techniques and baseline due to different access to test input/output pairs to decide correctness of code; thus, the reported improvement number on top of baseline or existing work is meaningless.

More concretely: the paper relies on an oracle to decide whether the code is correct or not to proceed into repair round. And this information is not part of the input from reranking / LEVER / Baseline / Reviewer (note: these baselines used test correctness at training time, but not in the evaluation inference time). As such, the self-repair framework has additional information gain to improve its performance (e.g., even if self repair only correct 1/100 case, it will be a straight improvement over the baseline). This different setup from existing work should be clearly mentioned, especially in the evaluation session; other readers could be confused on the reported "accuracy up to 12%" as claimed in the abstract.

Because I agree this paper is valuable to our community, I would like to propose two solutions to address this issue:
1. In self-repair, instead of using test I/O to decide if the code would be feed into the self-repair process, the authors can let the model to decide correctness of the code itself with only unit tests (for MBPP and transcoder) and task prompt. In this case, the model may make mistakes in classifying correct program as incorrect program, but it does not use the external power of test oracle. This approach would make the technique directly comparable with existing approaches and baseline no-repair model, because all of these approaches have same access of information when running evaluation.

2. Alternatively, modify the baseline to be able to access to the test oracle: when the baseline generate code that fails on test oracle, simply resample a solution. In this case, the baseline would also benefit from oracle, and the resample will be guaranteed to improve the test accuracy (but probably not as much as self-repair approach). This will also be a fair comparison and the numbers would be meaningful to the readers.

As of now, I don't think it is appropriate to accept the paper because the experiments are comparing apples to pineapples. But I would vote for an acceptance if this issue can be addressed given the paper is well written and innovative.

**Questions:**

Please address my concern on "comparison with baseline" clearly.

---

> ### Author Response · Authors · 2023-11-17
>
> Thanks for raising the concern. We appreciate that you consider our work as innovative and well-written. We would like to clarify our experimental setup as follows.
>
> ### Self-debugging does not have different test input/output pairs to decide the code correctness compared to baselines ###
>
> We want to emphasize that self-debugging does not utilize extra input-output pairs compared to the baselines. Specifically, for text-to-SQL generation, there is no unit test for both self-debugging and all other baselines. Therefore, the LLM needs to infer the code correctness completely by itself. As described in Section 4.3, on MBPP, we follow the baseline approaches to include 1 unit test in the problem description, and other hidden unit tests are only used for the final evaluation on the code correctness. Therefore, the LLM also needs to infer the code correctness, since passing the given unit test does not necessarily mean that the predicted code is fully accurate and can pass the hidden unit tests. Note that during the self-debugging process, the hidden unit tests are never used, and the generated code is only executed on the 1 given unit test that already shows up in the problem description for generating the initial code. For code translation, we consider the setting where all unit tests are available, since the ground truth execution results can already be obtained by executing the code in the source programming language, which also follows the evaluation setup in prior work.
>
> ### We compared to baselines that utilize code execution ###
> As described in Section 2 and Section 5 (page 5), we compared self-debugging to baselines that generate multiple programs, and use code execution to select the final response.
>
> On the Spider benchmark for text-to-SQL generation, since the unit tests are not available, the baseline takes the majority vote of execution results as the final prediction. As shown in Figure 4 (a), self-debugging from greedy decoding matches the performance of the execution-based baseline that uses 16 samples, and self-debugging consistently improves the sample efficiency. For Transcoder, our baseline utilizes all unit tests to determine the code correctness. Note that the performance of this baseline is equivalent to the reviewer’s second suggested solution, since we consider the prediction from this baseline to be accurate if any one of the generated samples is correct. Again, we show that self-debugging from greedy decoding matches the baseline that generates 10 samples, and we presented the full analysis in Appendix E.1.
>
> ### The usage of richer execution information is a strength of our self-debugging design ###
>
> First, we would like to note that the reranking techniques LEVER and MBR-Exec also utilize code execution to select the response. The main difference between our work and these execution-based reranking techniques lies in the way we utilize the execution results. Specifically, the baseline approaches generate multiple samples all at once, thus the model can generate the same wrong code multiple times despite observing the execution errors. On the other hand, self-debugging can leverage past predictions and code execution to reduce the redundant cost of resampling wrong predictions, which is an advantage over existing reranking techniques.
>
> Furthermore, the key motivation of our self-debugging design is to provide complementary performance gain on top of all kinds of existing code generation methods. As shown in Figure 1 and described in Section 3, the first step of each self-debugging turn is code generation, which can be implemented as any existing baseline such as greedy decoding and execution-based reranking. Therefore, the subsequent debugging step leverages richer execution information than the first step of code generation by design. We demonstrate in Table 1, Figure 4 (a) and Appendix E.1 that self-debugging does not only improve over greedy decoding, but also consistently improves the performance when the baseline utilizes multiple samples.
>
> We hope the above discussion addresses your concerns. Please do not hesitate to contact us for any further questions or clarifications.

---

> > ### Comment · Reviewer_AdYB · 2023-11-17
> > **Great response!**
> >
> > Thanks a lot for the clarification. It's indeed my misunderstanding about whether evaluation test cases are used in deciding whether a solution would be repaired or not.
> >
> > Just a little quick follow up:
> >
> > 1. In SQL case, since it is LLM's job to decide whether a solution is correct or not to proceed to self-debug phase, (1) what percentage of *initially correct solutions* are determined as "incorrect" and proceeded to the self-debug phase, and (2) whether these solutions are self-debugged into a wrong solution or remains correct?
> > (I ask this because I guess that unlike cases with unit tests, LLM may not have 100% accuracy of deciding whether a solution is correct so it may fix a small amount of correct solutions into wrong solutions)
> > Btw, I don't have any further question on SQL dataset, since it quite clear that self-debugging over-performs the results of majority voting based on 16 solutions, given that self-debugging only draws 2 samples (1 greedy + 1 self debug output). [Can you confirm if my understanding is right?]
> >
> > 2. For MBPP and transcoder, despite (greedy) baseline models have access to unit tests in the same way as self-debug, they don't test it on them. I'm wondering if it is possible to provide a quick enhanced baseline: when a initially sampled program p is incorrect (when evaluated on unit tests), resample a second program and use it towards calculating accuracy.
> > (I ask this because it would interesting to understand how much gain in self-debug comes from "having a second chance" v.s. "the generated self-debug feedback". This enhanced baseline is better than the current baseline because it only has one chance to generate the final solution without access to evaluation on unit tests.) Further clarification:
> >
> > def current_baseline(prompt, test_case):
> >    p = LLM(prompt)
> >    return p
> >
> > def enhanced_baseline(prompt, test_case):
> >    p = LLM(prompt)
> >    if p passes test_case: return p
> >    else: return LLM(prompt)
> >
> > def self_debug(prompt, test_case):
> >   p = LLM(prompt)
> >   if p passes test_case: return p
> >   else: return LLM_with_self_debug(prompt, p, test_case)
> >
> > I'm wondering if the authors agree that enhanced_baseline may be a better baseline to highlight the effect of self debug feedback? Or, did I mistake again here that enhanced_baseline is already the baseline presented in table 2?
> >
> > Besides, I fully agree that the way how unit tests are used is an important contribution of the paper. Thus I'd like to understand the above two points. I will update my score since SQL cases clearly show an improvement.

---

> > > ### Author Response · Authors · 2023-11-18
> > > **Thanks for acknowledging our response**
> > >
> > > Thanks for acknowledging our response and being willing to update your score! We are glad that our response addressed your concerns, and we answer your followup questions below.
> > >
> > > 1. For text-to-SQL generation, you are correct that self-debugging only draws 2 samples (1 greedy + 1 self debug output), while outperforming the majority voting based on 16 solutions. Also, LLM indeed might wrongly infer the solution correctness and turn correct solutions into wrong ones. Below are the statistics using Codex:
> > > * LLM wrongly inferred correct initial correct solutions as incorrect: 3.3%
> > > * LLM wrongly changed initial correct predictions into incorrect ones: 1.0%
> > > * LLM fixed initial wrong predictions into correct ones: 4.3%
> > >
> > > We observe that even if the LLM does not have 100% accuracy of deciding whether a solution is correct, the majority of modified solutions upon the initial correct solutions is still accurate, thus most incorrect predictions of the code correctness do not directly lead to performance degradation.
> > >
> > > 2. Thanks for suggesting the enhanced baseline! You are right that the baseline in Table 2 does not have a chance to generate another solution when the greedy decoding produces the wrong code. We evaluated the enhanced baseline using Codex, and the accuracies are as follows:
> > > * Transcoder: greedy decoding = 80.4%, enhanced baseline = 84.5%, self-debugging = 92.5%
> > > * MBPP: greedy decoding = 61.4%, enhanced baseline = 65.4%, self-debugging = 70.8%
> > >
> > > These results show that self-debugging outperforms the enhanced baseline, which highlights the effect of self-debugging feedback.
> > >
> > > Note that the design of the baseline with multiple samples in Table 1 in the main paper and Figure 5 (b) in Appendix E.1 is equivalent to the enhanced baseline, except that these results were obtained with more than two trials. For example, Figure 5 (b) shows that when the baseline uses 5 samples (equivalent to the enhanced baseline with 5 trials), the accuracy is 90.9%, which is still worse than self-debugging that reaches 92.5% accuracy.
> > >
> > > We hope these results address your questions. Please let us know your thoughts, and we are more than happy to answer any further questions.

---

> > > > ### Comment · Reviewer_AdYB · 2023-11-20
> > > >
> > > > Thanks a lot for clarification, I'm satisfied with explanation on Spider dataset, and hopefully these discussions and results will be integrated into the paper (updated my score based on this).
> > > >
> > > > For this one:
> > > >
> > > > Transcoder: greedy decoding = 80.4%, enhanced baseline = 84.5%, self-debugging = 92.5%
> > > > MBPP: greedy decoding = 61.4%, enhanced baseline = 65.4%, self-debugging = 70.8%
> > > >
> > > > 80.4% seems to be Codex performance on baseline Transcoder, and self-debugging = 92.5% seems to be the results from GPT-3.5. What's the enhanced baseline number based upon? I'm wondering if it is possible to add enhanced baseline result into Table 2 for easy comparison (for both MBPP and Transcoder)?

---

> > > > > ### Author Response · Authors · 2023-11-21
> > > > > **We added the enhanced baseline results with all LLMs in Table 2 of the paper**
> > > > >
> > > > > Thanks for updating the score!
> > > > >
> > > > > First, we would like to clarify that all results below used Codex:
> > > > >
> > > > >    - Transcoder: greedy decoding = 80.4%, enhanced baseline = 84.5%, self-debugging = 92.5%
> > > > >    - MBPP: greedy decoding = 61.4%, enhanced baseline = 65.4%, self-debugging = 70.8%
> > > > >
> > > > > The self-debugging numbers are the bolded Codex numbers in Table 2b and 2c.
> > > > >
> > > > > Based on your suggestion, we evaluated all LLMs with the enhanced baseline on Transcoder and MBPP. We revised the paper to add these results (denoted as the oracle baseline) in Table 2, added the description of this baseline in the first paragraph of Section 5 (page 5), and added discussion of these results in Section 5.1. Basically, the oracle baseline results with different LLMs further justify that self-debugging is more effective than completely discarding wrong predictions and sampling another program independently. Note that on MBPP, the oracle baseline has access to all unit tests to determine the code correctness, while self-debugging only has access to 1 unit test. Nevertheless, the oracle baseline still generally performs worse than self-debugging with the simple feedback, let alone other feedback types.
> > > > >
> > > > > We hope the new results with paper revision further address your concerns. Thanks for acknowledging that our paper is well-written and our approach is innovative, and we are more than happy to answer any followup questions you may have to support the acceptance of our work.

---

> > > > > > ### Comment · Reviewer_AdYB · 2023-11-21
> > > > > >
> > > > > > Thanks a lot. The revision addresses my concern. I will further update my score.

---

> > > > > > > ### Author Response · Authors · 2023-11-22
> > > > > > > **Thanks for your update!**
> > > > > > >
> > > > > > > Thanks for your support! We are glad that our response addressed your concerns, and thank you again for your constructive feedback.

---

### Official Review · Reviewer_UhXy · 2023-10-29

**Soundness:** 3 good
**Presentation:** 3 good
**Contribution:** 3 good
**Rating:** 6
**Confidence:** 4

**Summary:**

This paper propose a method called self-debugging, which teaches a large language model to debug its predicted program via few-shot demonstrations.The intuition is that human programmers also write programs in an iterative way where they modify the code through debugging. Their method achieves state-of-the-art performance on several code generation benchmark.

**Strengths:**

The intuition and introduction of this paper is quite clear. The proposed method is simple, effective, and can be applied to any LLM model. Their method achieves promising performance improvement in code generation tasks.

**Weaknesses:**

1. Although the introduction of this paper is clear, the methodology part is not the case. There are many components in the proposed approach: code execution, code explanation, inferring code correctness, etc. Figure 1 is helpful but still not clear enough. It would be better if there is a diagram with concrete example in the main text.
2. Although the paper claims that they improve sample efficiency, I am still doubtful about this as the self-debugging approach does require generating much more tokens for debugging. I think comparing different approaches by the average the number of tokens generated for solving each problem is more fair.

**Questions:**

1. How do you compare your approach to Reflexion [1]?
2. The paper has shown that self-debugging is effective in fixing grammar and semantic error. Is it effective in fixing logic error?  For example, the original implementation is correct but does not actually solve the problem. Will self-debugging be able to identify and fix such type of error?

[1] Noah Shinn, Beck Labash, and Ashwin Gopinath. Reflexion: an autonomous agent with dynamic memory and self-reflection. arXiv preprint arXiv:2303.11366, 2023.

---

> ### Author Response · Authors · 2023-11-17
>
> We thank the reviewer for the encouraging comments.
>
> ### Concrete example of self-debugging ###
>
> Thanks for the feedback. We have added Figure 2 and 3 in the main paper to show the self-debugging workflow, which are abbreviated versions of Figures 6 and 7 in Appendix F. The 2 figures demonstrate concrete running examples of how self-debugging works with different prompt formats.
>
> ### Sampling budget comparison ###
>
> Compared to the sampling cost for obtaining the initial code for debugging, self-debugging requires the LLM to generate more tokens by design. For example, in Table 2, self-debugging generates more samples than the greedy decoding baseline. This extra cost is also needed for other related work, e.g., self-refine and Reflexion.
>
> On the other hand, as discussed in Section 5.2.1, we demonstrate that to improve the accuracy over greedy decoding, self-debugging achieves better performance with a lower inference cost. To compare the cost in terms of the number of tokens, we consider two settings: (1) when unit tests are not available; e.g., on Spider for text-to-SQL generation; and (2) when unit tests are available; i.e., on Transcoder and MBPP for Python code generation.
>
> On the Spider benchmark, with explanation feedback, on average a full debugging turn generates 5.5 times more tokens than simply generating the SQL query. By comparing the performance of self-debugging starting from greedy decoding (80.8% accuracy) to the baseline that samples 16 programs without self-debugging (80.7% accuracy), we can see that self-debugging also reduces the number of generated tokens compared to the baseline.
>
> On Transcoder and MBPP, due to the availability of unit tests, the improvement with explanation feedback is generally less significant than on Spider. In this case, the unit test feedback is a better option to reduce the sampling budget. On average, a full debugging turn with the unit test feedback generates 2.5 times more tokens than simply generating the initial code, thus self-debugging is again more sample-efficient in terms of the token cost.
>
> Furthermore, despite the fact that generating explanation and trace feedback increases the number of generated tokens, we note that one benefit of such feedback is to provide more interpretable rationales on how LLMs infer the code correctness, and how the fixes are conducted. Such natural language feedback helps us to investigate the models’ weaknesses, and potentially can be used to finetune the model and improve its overall debugging and coding ability.
>
> ### Comparison to Reflexion ###
> The Reflexion paper evaluates their approach on AlfWorld and HotPotQA, which are decision-making tasks and question-answering tasks. For both benchmarks, they assume that the oracle reward is available, which tells the LLM whether the current state is successful or the current answer is correct.
>
> On the contrary, our work focuses on code generation tasks, where the LLM does not always have access to the oracle reward about the code correctness. For example, on text-to-SQL generation where the unit tests are not available, the LLM needs to determine completely by itself when the debugging process should terminate. Furthermore, we proposed different prompt formats to enable self-debugging for diverse tasks and LLMs, and we demonstrated that the self-debugging ability can be taught with few-shot prompting, even if the model is not specially tuned for zero-shot debugging. For example, with code-davinci-002 that is not instruction tuned, we showed that self-debugging with few-shot prompting achieves similar or better performance gain than GPT-4.
>
> ### Questions on fixing logic errors ###
>
> For all tasks, our evaluation metric is the execution accuracy, which considers the predicted code to be correct when it passes the given unit tests or its outputs match those of the ground truth code. Therefore, when we only use unit tests to determine whether to continue the self-debugging process, e.g., for Transcoder experiments where all unit tests are available, programs that pass the unit tests but have other logic errors would not be detected. On the other hand, when we request the LLM to determine the correctness of the generated code, e.g., on Spider and MBPP, the LLM can correct such logical errors to some extent. In our experiments, we manually investigated tens of correct predictions from each benchmark, and we did not notice such false positives.

---

> > ### Comment · Reviewer_UhXy · 2023-11-22
> > **Response to rebuttal**
> >
> > Thanks for your response to my questions. I would like to keep my positive evaluation.

---

> > > ### Author Response · Authors · 2023-11-22
> > > **Thanks**
> > >
> > > Thanks for supporting our work!

---

### Official Review · Reviewer_hHB2 · 2023-10-31

**Soundness:** 3 good
**Presentation:** 3 good
**Contribution:** 2 fair
**Rating:** 6
**Confidence:** 4

**Summary:**

Authors propose Self-Debugging approach in which LLMs identify code generation mistakes by investigating the execution results and explaining the generated code. This approach can improve code generation quality. Authors have performed experiments on several code generation benchmarks.

**Strengths:**

- Proposed Self-Debugging approach based on code self-explanation ( rubber duck debugging ) and test execution result investigation.
- Some improvement over baseline results.

**Weaknesses:**

- Improvement over baseline for Spider benchmark is only 2-3% which is not shown to be statistically significant. It could be accidental result of prompt change.
- Same issue with code explanation without debugging for TransCoder and MBPP.
- It seems that Self-Debugging without unit tests executions has very limited and possibly statistically insignificant improvements.

The following weaknesses have been fixed in the paper update by the authors:
- Section 4 is very hard to read. It constantly refers to Appendixes. Even with limited space, it is possible to present material much better so that paper would be self-contained and would not require readers to read appendixes to follow a whole section 4. In my opinion, if this is not fixed, this is a very serious issue with the paper.
- There are presentation issues with result tables. See Questions.

I have increased my rating from 3 to 5.
Based on further responses from the authors, I increased my rating to 6.

**Questions:**

- In table 1, what does line "Codex" represent? It is in the section "Self-Debugging (this work)", but the result is different from other lines in Self-Debugging section. Your paper does not explain what this line shows. Is this baseline without Self-Debugging techniques? If so, why is it presented in Self-Debugging section?

- In table 2, Codex results for Spider correspond to Codex results for Spider in Table 1. However, Codex results for MBPP do not correspond to Codex results for MBPP in Table 1. Why?

---

> ### Author Response · Authors · 2023-11-17
>
> We thank the reviewer for the constructive feedback.
>
> ### Significance of the Spider improvement ###
>
> We would like to note that 2-3% improvement on Spider is non-trivial. As shown in Figure 4 (a), without self-debugging, a 3% improvement from greedy decoding requires 16 samples. In particular, Figure 4 (b) shows that self-debugging improves the accuracy on the hardest SQL tasks by 9%, which is a significant boost.
>
> ### Significance of the code explanation on Transcoder and MBPP ###
>
> First, we would like to clarify that code explanation results in Tables 1-3 are accompanied with self-debugging. The evaluation of code explanation without self-debugging is not our primary focus, but we presented the results in Figure 1 of Appendix E.1, where code explanation consistently improves the performance by 2-4%.
>
> On the other hand, due to the availability of unit tests, we acknowledge that the improvement with explanation feedback on Trancoder and MBPP is generally less significant than on Spider. However, one benefit of explanation feedback is to provide more interpretable rationales on how LLMs infer the code correctness, and how the fixes are conducted. Such natural language feedback helps us to investigate the models’ weaknesses, and potentially can be used to finetune the model and improve its overall debugging and coding ability.
>
> Furthermore, we would like to emphasize that one core contribution of our work is to conduct a comprehensive evaluation of self-debugging performance with different LLMs, applications, prompting methods and feedback formats, and demonstrate the importance of each factor. Specifically, both code explanations and traces are also commonly used for human debugging, which do not require any extra input from other human programmers. We show that stronger LLMs for code (code-davinci-002 with few-shot prompting and gpt-4) benefit more from such richer self-generated feedback, which can motivate future work on improving LLMs for investigating and understanding the semantic meaning of code.
>
> ### Importance of unit test execution ###
>
> As discussed in Section 5.2.2, we acknowledge that the improvement of self-debugging is less significant without unit test execution. However, even without unit tests, sometimes LLMs can still improve their performance with self-generated feedback. For example, as shown in Table 3b:
> * With code-davinci-002, self-debugging without unit test execution improves the performance by up to 5%;
> * GPT-4 without unit test execution improves the MBPP accuracy by 3.6%.
>
> Furthermore, we would like to note that utilizing unit tests whenever applicable is also part of our self-debugging approach, and we consider the ability of leveraging execution of past generated programs as a key advantage of our self-debugging approach.
>
> ### Comments on paper writing ###
> Thanks for the feedback about Section 4’s readability. We have added Figure 2 and 3 in the main paper to show the self-debugging prompt formats, which are abbreviated versions of the corresponding figures in the appendix. We hope this makes the section easier to read.
>
> Regarding other questions:
>
> * Codex in Table 1 represents the baseline without self-debugging. We put it under the section “Self-Debugging (this work)” to denote that the following self-debugging results are built upon initial programs generated from this baseline. We added a divider around “Codex (baseline)” in Table 1 to make it clearer that these are the baseline results before self-debugging.
> * The Codex results in Table 2 were produced from greedy decoding, which demonstrate that self-debugging improves the performance with a higher sample efficiency. On the other hand, Codex baseline results in Table 1 were obtained with multiple samples, which demonstrate that self-debugging outperforms other baseline methods that leverage multiple samples. We updated the captions of Table 1 and 2 to clarify the settings.
>
> We hope our paper revision and above discussion address your concerns. Please do not hesitate to contact us for any further questions or clarifications.

---

> > ### Comment · Reviewer_hHB2 · 2023-11-19
> > **Thanks to authors for answers and paper changes**
> >
> > Thanks to authors for answers and paper changes.
> > I have updated my review and paper score based on paper changes.

---

> > > ### Author Response · Authors · 2023-11-19
> > > **Followup to address your remaining concerns**
> > >
> > > Thanks for acknowledging our response and updating the score! We are glad that our revision addressed your concerns on paper writing.
> > >
> > > We would like to follow up and further address your concerns on the significance of the results. To provide more evidence that it is non-trivial to improve the performance by 2-3% over the strong baseline, we conducted an evaluation on Spider benchmark with code-davinci-002, where the model generates 100 SQL queries for each question without self-debugging. With execution-based code selection, the accuracy is 82.7%, which improves over the baseline using 32 samples without self-debugging by 1.4%. This result is still slightly worse than self-debugging from 8 samples (82.9%). In particular, with 9% improvement on the hardest SQL tasks, as well as ~10% improvement on Transcoder and MBPP, we show that self-debugging significantly boosts the performance. Different feedback formats and whether or not unit tests are accessible lead to different degrees of performance improvement, demonstrating the effect of each component in the self-debugging scheme.
> > >
> > > We hope the above discussion further addresses your concerns. Please let us know your thoughts, and we are more than happy to answer any further questions.

---

> ### Comment · Reviewer_hHB2 · 2023-11-19
> **2-3% improvement significance**
>
> I think you may have misunderstood my comment regarding 2-3% improvements. What I am wondering about is whether simple prompt changes may result in improvements of similar range.
>
> We know that prompt changes affect results.
> We don't know if baselines and self-debugging use best possible prompts, so-so prompts, or not very good prompts. It is possible that the result ranges of baseline and self-debugging overlap if the prompt ranges are considered. We don't know how the results would compare if the best "oracle" prompts would be used.
>
> Determining reasonable prompt variability range may not be simple. Authors naturally pick the best prompt that they have seen so far. Different authors may spend less or more effort to find the best prompt.
>
> Have you tried different prompts for self-debugging? What variability range did you see?
>
> Thank you

---

> ### Author Response · Authors · 2023-11-20
> **Improvement from self-debugging is complementary to the improvement from baseline prompt changes**
>
> Thanks for your followup response! We agree that changing the baseline prompt can also change the performance. However, we have done our best to optimize the prompts and construct strong baselines. Please let us know if you have any better baseline prompts in mind, and we are more than happy to evaluate them.
>
> On the other hand, the improvement from self-debugging is complementary to the improvement from baseline prompt changes. As supporting evidence, for text-to-SQL generation, we evaluated a variant of our full prompt where we removed the demonstration of one data row per table; i.e., we deleted the INSERT INTO statements in Appendix H. With greedy decoding, the accuracy of this baseline prompt is 75.6%, which is 1.9% lower than our full prompt. From this baseline, self-debugging also improves the accuracy by 2.1%. Note that removing INSERT INTO statements weakens both the baseline and the self-debugging prompt, but we still see a consistent improvement from the self-debugging process.
>
> Regarding different self-debugging prompts, we have shown results of integrating different feedback information in Section 5, especially Table 2. During the development of our approach, we have also done extensive evaluation on different wordings of each feedback type, but we did not observe much performance differences that change the relative order when comparing different feedback types.
>
> We hope the above discussion and followup experiments address your concerns. Please let us know your thoughts, and we are more than happy to answer any further questions.

---

> > ### Author Response · Authors · 2023-11-22
> > **Follow-up**
> >
> > Hi Reviewer hHB2,
> >
> > We want to follow up to check if you had a chance to read our latest response. We would love to know if our response addressed your concerns about the significance of our results, and we are happy to continue the discussion and answer any further questions you may have.

---

> > > ### Comment · Reviewer_hHB2 · 2023-11-22
> > > **Thank you for authors response**
> > >
> > > Thank you for your follow up. It addresses my concerns about the results demonstrated in the paper.

---

> > > > ### Author Response · Authors · 2023-11-22
> > > >
> > > > We are happy to hear that!
> > > > In light of our addressing your concerns, would you be open to re-considering the rating for our paper?

---

### Official Review · Reviewer_urzn · 2023-11-01

**Soundness:** 3 good
**Presentation:** 3 good
**Contribution:** 3 good
**Rating:** 6
**Confidence:** 3

**Summary:**

This work proposes a novel approach called SELF-DEBUGGING that enables a large language model to debug its own predicted program via few-shot demonstrations. The approach achieves state-of-the-art performance on several code generation benchmarks, including the Spider dataset for text-to-SQL generation, TransCoder for C++-to-Python translation, and MBPP for text-to-Python generation. The paper also discusses the possibilities of leveraging feedback messages and reusing failed predictions to improve the sample efficiency.

**Strengths:**

1. Proposing a novel approach called SELF-DEBUGGING that enables a large language model to debug its own predicted program via few-shot demonstrations.
2. Demonstrating that SELF-DEBUGGING can teach the large language model to perform rubber duck debugging, i.e., identifying its mistakes by investigating the execution results and explaining the generated code in natural language.
3. Achieving state-of-the-art performance on several code generation benchmarks, including the Spider dataset for text-to-SQL generation, TransCoder for C++-to-Python translation, and MBPP for text-to-Python generation.
4. Discuss the ethics of using large language models for automated code generation and the importance of analyzing their capabilities and limitations before deploying them for real-world programming applications.

**Weaknesses:**

I do not find obvious weaknesses in this work. I only have a concern about the proposed approach. Since I'm not an expert in the code generation field, please correct me if I have some misunderstandings. This kind of "self-debug" or "self-refine" requires LLMs to inspect their own outputs based on the unit test results and generate some explanation in an autoregressive manner. So a concern is the additional latency in the inference time, especially for extremely large language models. This extra computational time is not reported or discussed in this paper.

**Questions:**

None

---

> ### Public Comment · ~Dong_HUANG4 · 2023-11-13
> **Self-debug is not new**
>
> For the overhead of the self-debug, I believe that we can simply consider each interaction with LLM can increase one running time for code generation. However several researchers have pointed out this overhead is worthwhile since it can largely increase code generation effectiveness. While I think the key problem of this paper is that Self-Debug is not new for code generation, previous strategies, e.g., Self-Edit, Self-Refine, and Self-Collaboration are used for code generation, but the authors have not compared with even one baseline in their experiments?

---

> > ### Comment · Reviewer_urzn · 2023-11-13
> >
> > Hi thanks for the insightful comments.
> >
> > I've checked the publication date of this paper and Self-refine on arxiv and find that they are almost in the same month. So I consider them as concurrent work, and both propose the "self-eval, self-refine, self-debug" idea. I agree that the authors can compare with these baselines to make the results more solid.
> >
> > For the overhead of this approach, maybe quantitative results, quantifying the increase in both overhead and the overall performance, are needed to show whether this overhead is really worthwhile.

---

> > > ### Author Response · Authors · 2023-11-17
> > >
> > > We thank the reviewer for the encouraging comments.
> > >
> > > ### Questions on extra compute ###
> > > Regarding the question on the extra computation time: compared to the sampling cost to obtain the initial code for debugging, self-debugging requires extra compute by design. For example, in Table 2, self-debugging generates more samples than the greedy decoding baseline. This extra computation is also needed for other related work, e.g., self-refine and Reflexion.
> > >
> > > On the other hand, as discussed in Section 5.2.1, compared to the baselines that generate multiple samples from scratch, self-debugging achieves better performance with a lower inference cost. For example, as shown in Figure 4 (a), for text-to-SQL generation, to achieve similar performance as self-debugging, the baseline requires 16 samples, which means running forward passes 16 times. Similarly, as shown in Appendix E.1, self-debugging from greedy decoding achieves comparable performance to the baseline with 10 samples for code translation. Given that self-debugging with one turn already brings most of the performance gain (as discussed in Section 5.2.1), these results show that to improve the LLM coding performance on top of greedy decoding, self-debugging is more cost-effective than baseline methods that don’t have debugging steps.
> > >
> > > ### Comparison to Self-Refine ###
> > >
> > > Thanks for your response acknowledging that our work is concurrent with self-refine, and our work was released on arxiv earlier than the Self-Edit and Self-Collaboration papers mentioned in the public comment. As self-refine did not evaluate on code generation tasks, their prompts are not directly applicable to our benchmarks, and they did not design feedback formats tailored to coding such as code execution and explanations. Specifically, self-refine utilizes GPT-3.5 and GPT-4 to generate feedback to improve its performance on several tasks, where they mainly show performance improvement on text generation tasks that do not require complex reasoning. On the other hand, self-debugging focuses on code generation, and we present a comprehensive study on different feedback formats across different coding benchmarks.
> > >
> > > In particular, one key difference between self-debugging and other concurrent and followup work on using LLM-generated feedback to improve test-time performance is we demonstrate that the self-debugging ability can be taught with few-shot prompting, even if the model is not specially tuned for zero-shot debugging. Specifically, with code-davinci-002 that is not instruction tuned, we show that few-shot prompting achieves similar performance gains as GPT-3.5 and GPT-4, while its self-debugging performance is much better for text-to-SQL generation than GPT-3.5 and GPT-4.

---

> > ### Public Comment · ~Yixuan_Weng1 · 2023-11-23
> >
> > Perhaps self-verification came a bit earlier...it was posted on arxiv in December 2022.
> >
> > Of course, it was not designed specifically for code.
> >
> > [1] Weng Y, Zhu M, He S, et al. Large language models are reasoners with self-verification[J]. arXiv preprint arXiv:2212.09561, 2022.

---

> > > ### Author Response · Authors · 2023-11-23
> > >
> > > Thanks for sharing this interesting work. We would like to note two major differences between self-verification and our work:
> > >
> > > 1. Self-verification includes a backward verification process to compute verification scores after the model generates multiple candidate answers, and there is no response improvement process afterward. On the other hand, self-debugging adds a debugging step after the LLM generates the initial code, which can improve upon greedy decoding.
> > >
> > > 2. Self-verification was designed for reasoning tasks including arithmetic reasoning, commonsense reasoning and logical reasoning, and the improvement on arithmetic reasoning is more significant than other tasks. On the other hand, as also noted in the comment, our self-debugging work focuses on code generation.
> > >
> > > We have updated our paper to add the discussion of this work.

---

### Author Response · Authors · 2023-11-17
**Summary of paper revision**

We thank all reviewers for the constructive comments. We appreciate the reviewers’ positive feedback that our approach is novel and effective across tasks and models. Based on reviewers’ suggestions, we uploaded a revision of our paper with the following major changes:
* Based on Reviewer hHB2 and UhXy’s comments, we have added Figures 2 and 3 in the main paper to show the self-debugging prompt formats, which are abbreviated versions of Figures 6 and 7 in Appendix F. The 2 figures demonstrate concrete running examples of how self-debugging works.
* Based on hHB2’s comments, we have updated the captions of Table 1 and 2 to clarify the settings. Specifically, Table 1 presented self-debugging results on Codex with multiple samples, which demonstrates that self-debugging outperforms existing reranking methods that utilize multiple samples. Table 2 presents self-debugging results starting from greedy decoding, which demonstrates that self-debugging consistently improves the performance across different LLMs, and compares the performance of different prompt formats.
* We revised Section 6 to highlight the differences between our approach and related work such as Reflexion and Self-Refine. Specifically, Reflexion focuses on the setting where the oracle reward is available, which tells the LLM whether the current state is successful or the current answer is correct. Self-refine utilizes GPT-3.5 and GPT-4 to generate feedback to improve its performance on several tasks, where they mainly show performance improvement on text generation tasks that do not require complex reasoning. On the other hand, our work focuses on code generation tasks, where the LLM does not always have access to the oracle reward about the code correctness. For example, on text-to-SQL generation where the unit tests are not available, the LLM needs to determine completely by itself when the debugging process should terminate. Furthermore, we propose different prompt formats to enable self-debugging for diverse tasks and LLMs, and we demonstrate that the self-debugging ability can be taught with few-shot prompting, even if the model is not specially tuned for zero-shot debugging. For example, with code-davinci-002 that is not instruction tuned, we show that self-debugging with few-shot prompting achieves similar or better performance gain than GPT-4.

We believe we have addressed each reviewer’s concerns and questions in the individual responses. Please do not hesitate to contact us for any further questions or clarifications.

---

### Meta-Review · Area_Chair_p125 · 2023-12-04

**Metareview:**

This paper proposes a self-debugging method, which enables a large language model to debug its predicted program without any human feedback on the code's correctness or error messages.

A reasonable amount of discussions took place between the authors and the reviewers and among the reviewers themselves. In the end, we got four reviews with ratings of 6, 6, 6, and 6 with confidence of 3, 4, 4, and 5 respectively. The reviewers appreciate the novel framework and the comprehensive experiments.

Reviewers proposed issues about computational cost (urzn), presentation (hHB2), the effectiveness of the method (UhXy), and the unfair comparison in the experiments (AdYB). Fortunately, the authors have addressed the main issues proposed by the reviewers (hHB2, UhXy, AdYB).

**Justification For Why Not Higher Score:**

The paper is considered good but there is also concurrent works exist.

**Justification For Why Not Lower Score:**

Most concerns that were raised have been well-addressed by the authors and the reviewers are consensus to be accepted with score at least 6.

---

### Decision · Program_Chairs · 2024-01-16

Accept (poster)